# Exploring the potential of dental calculus to shed light on past human migrations in Oceania

Irina M. Velsko [1], Zandra Fagernäs [1,24], Monica Tromp [2,3,4], Stuart Bedford[5,6], Hallie R. Buckley [4], Geoffrey Clark [5], John Dudgeon[7], James Flexner[8], Jean-Christophe Galipaud[9], Rebecca Kinaston [10], Cecil M. Lewis Jr [11], Elizabeth Matisoo-Smith [4], Kathrin Nägele [1], Andrew T. Ozga [12], Cosimo Posth [1,13,14], Adam B. Rohrlach [1,15], Richard Shing[16], Truman Simanjuntak[17], Matthew Spriggs [16,18], Anatauarii Tamarii[19], Frédérique Valentin [20], Edson Willie[16] & Christina Warinner [1,21,22,23] ✉

The Pacific islands and Island Southeast Asia have experienced multiple waves of human migrations, providing a case study for exploring the potential of ancient microbiomes to study human migration. We perform a metagenomic study of archaeological dental calculus from 102 individuals, originating from 10 Pacific islands and 1 island in Island Southeast Asia spanning ~3000 years. Oral microbiome DNA preservation in calculus is far higher than that of human DNA in archaeological bone, and comparable to that of calculus from temperate regions. Oral microbial community composition is minimally driven by time period and geography in Pacific and Island Southeast Asia calculus, but is found to be distinctive compared to calculus from Europe, Africa, and Asia. Phylogenies of individual bacterial species in Pacific and Island Southeast Asia calculus reflect geography. Archaeological dental calculus shows good preservation in tropical regions and the potential to yield information about past human migrations, complementing studies of the human genome.

Archaeogenetics studies of ancient human migrations are conventionally conducted by analysing human DNA from skeletal elements, such as teeth and the petrous portion of the temporal bone. Human population movements across the Pacific and Island Southeast Asia (ISEA) have been reconstructed in this manner, revealing successive waves of migration during the Pleistocene and Holocene[1–6]. However, ancient DNA work in the Pacific and ISEA is challenging as the studied populations are closely related and migrations take place over short periods of time. Further, the high temperatures and humidity in the tropics generally increase the rate of DNA decay[7,8], making human DNA preservation highly variable[1]. Ancient dental calculus, the calcified oral biofilm preserved on teeth, offers a possible alternative approach to studying human migrations in the Pacific and culturally associated sites in ISEA, which has not yet been widely explored[9,10].

Dental calculus forms when the bacterial biofilm known as dental plaque naturally calcifies on the surfaces of teeth. The dental plaque microbes become encased within the mineral matrix, preserving biomolecules including DNA, proteins, and small molecule metabolites, and this enables studies of oral bacterial communities stretching thousands of years back in time[11]. Because microbes have shorter generation times than humans, they offer the possibility to study the migrations of closely related populations over shorter timescales. Further, DNA within dental calculus is generally better preserved than DNA in skeletal tissues from the same individual[12], making it a promising substrate to study in areas where skeletal DNA preservation is generally poor, such as the tropics.

The prospect of studying human migration through archaeological dental calculus presents several potential advantages, as it would allow

for a holistic study of an individual's life through just one sample - from migration and diet, to health and disease, and even occupational activities[13]. Dental calculus may prove to be an especially valuable study material in the Pacific, given the speed of settlement of the region and significant cultural changes over a short time. However, no study has to date attempted to investigate past human migrations through the oral microbiome, although the prospect has been discussed[9,10].

Here, we explore the possibility of using archaeological dental calculus to study past human migrations, using Pacific and ISEA islands as a case study. Shotgun metagenomic sequencing was performed on a total of 102 dental calculus samples from 10 Pacific islands and 1 island in ISEA, spanning a time range of nearly 3000 years (Table 1, Supplementary Data 1, 2). We show that DNA preservation is variable, but that most samples have a well-preserved oral microbiome. This highlights the exceptional preservation of DNA in dental calculus, even in challenging environments. We find that variation in dental calculus microbial community composition does not have a clear geographic or temporal structure, but rather may be influenced by local factors specific to each island. In contrast, phylogenetic analyses of individual oral bacterial taxa exhibit temporal trends, but the ability to detect this signal depends on the selected species' prevalence and abundance. Overall, we find that metagenomic analysis of archaeological dental calculus has the potential to reveal information about past human migration patterns. When combined with human DNA analysis and other approaches, such as paleodietary studies using palaeoproteomics and microremains, it promises to enrich our understanding of the dynamic biological and cultural processes that accompanied past migrations across ISEA and the Pacific.

## Results

### Preservation of endogenous ancient DNA

Obtaining well-preserved ancient DNA is a persistent challenge in archaeogenetic studies of the tropics, as both temperature and humidity contribute to DNA decay[7,8]. Using SourceTracker analysis[14], we estimated the proportion of microbial taxa originating from endogenous and contaminant sources for each dental calculus sample in this study (Fig. 1A). For 73 of the 102 archaeological dental calculus samples, at least 50% of microbial content is estimated to originate

from an oral microbiome source, indicating good preservation of the dental calculus for these samples, with minimal contamination from exogenous sources. We further assessed the proportion of endogenous oral species using cuperdec[11] (Supplementary Fig. 1) and found high consistency between estimated sample preservation using both methods. A PCA of well-preserved samples shows minimal overlap between dental calculus and the source samples used in SourceTracker, with calculus samples clustering distinctly from all sources (Fig. 1B). All samples that passed the cuperdec threshold for preservation, 73 samples, were carried forward for analysis (Fig. 1C).

The effect of sample age and environmental variables (average rainfall, average temperature, and average evapotranspiration of each island) on the proportion of taxa that could be assigned to oral sources (dental plaque and dental calculus), as well as the proportion of human DNA recovered from calculus, was investigated using beta regression (Fig. 1D). Only sample age was a significant predictor of preservation ($p = 0.046$, $R^2 = 0.083$), yet the low $R^2$ value indicates that additional, unknown factors more strongly affect preservation. Local environmental factors at the burial site and for each individual grave, such as soil type, humidity, and pH, are plausible candidates, but such data are not available.

Lastly, because the environment of the Pacific is less conducive to DNA preservation than colder, drier climates, we compared preservation patterns between the Pacific/ISEA and northern Europe, and we asked whether the relative preservation of endogenous oral microbiome DNA in dental calculus is greater than that of human DNA in archaeological bone/teeth (Fig. 1E). For the analysis, we considered all oral microbial DNA identified in dental calculus to be endogenous, and all human DNA present in bone/teeth to be endogenous. Overall, we find that endogenous preservation of ancient dental calculus in the Pacific/ISEA is high, in which 72/102 samples (70%) are estimated to derive 80% or more of their composition from an oral microbiome source. This is only somewhat lower than that estimated for dental calculus from northern Europe, where 106/112 samples (95%) are estimated to derive more than 80% of their composition from an oral microbiome source. In contrast, human DNA preservation in skeletal material was much lower for both the Pacific and northern Europe, with 5/183 (2.7%) and 19/464 (4.1%) samples having endogenous DNA content of at least 5%, respectively, for samples in which ancient human DNA was detected. These results suggest that DNA within calculus is less prone to degradation and exogenous contamination than DNA in archaeological skeletal remains and may be a more promising avenue of study for sites in warmer, more humid climates.

### Microbial community composition

We next asked whether the microbial community of well-preserved Pacific/ISEA calculus samples showed temporal or spatial differences. If present, such patterns might suggest that the calculus microbiome changed with human migration through ISEA and the Pacific, although specific factors affecting such changes would need elucidation. We performed a beta-diversity analysis and visualized the samples using PCA (Fig. 2A) and tested whether multiple clusters were present in the data. Testing the goodness of fit of sample cluster numbers to the data indicated that a single cluster optimally described the data (Supplementary Fig. 2). This corresponded with the visual lack of distinct sample clustering in the PCA plot, in which the samples did not tightly cluster based on island or time period, nor did they plot along a cline that might suggest temporal or geographic change. PERMANOVA determined that the processing lab ($R^2 = 0.0275$, F = 2.748, $p = 0.01$), island ($R^2 = 0.1913$, F = 1.736, $p = 0.001$), and average GC content ($R^2 = 0.0508$, F = 5.068, $p = 0.002$) were the most influential factors in how the samples plotted. When controlling for the lab in which samples were processed, island ($R^2 = 0.2045$, F = 1.650, $p = 0.035$) and average GC content ($R^2 = 0.0761$, F = 7.372, $p = 0.01$) were still significant drivers of the sample community composition.

## Table 1 | Archaeological dental calculus samples included in this study

| Island | Island group/Area | # Samples (# passed) | Age BP[a] | Processing lab |
|---|---|---|---|---|
| Efate | Vanuatu, Pacific | 5 (5) | 1050–0 | Jena |
| Efate | Vanuatu, Pacific | 16 (10) | 3000–2300 | Otago |
| Flores | Lesser Sunda Islands, ISEA | 3 (3) | 3000–2100 | Oklahoma |
| Futuna | Vanuatu, Pacific | 3 (2) | 1270–290 | Jena |
| Raiatea | Society Islands, Pacific | 2 (2) | 240–0 | Jena |
| Rapa Nui | Rapa Nui, Pacific | 18 (15) | 630–0 | Oklahoma |
| Taumako | Duff Islands, Pacific | 17 (16) | 440–0 | Jena |
| Tongatapu | Tonga, Pacific | 6 (5) | 2700–2400 | Jena |
| Uripiv | Vanuatu, Pacific | 6 (3) | 2600–2000 | Otago |
| Vao | Vanuatu, Pacific | 5 (4) | 2750–2000 | Otago |
| Viti Levu | Fiji, Pacific | 6 + 13 (4 + 2)[b] | 1700–1300 | Oklahoma +Jena |
| Watom | Bismarck Archipelago, Pacific | 2 (1) | 2700–2300 | Otago |

[a]Age BP is the estimated age range for the individuals by site. See Supplementary Data 1 for details.

[b]Samples from Viti Levu were extracted in two labs, either in Oklahoma or in Jena. Of the 6 samples extracted in Oklahoma, 4 were well-preserved, while of the 13 extracted in Jena, only 2 were well-preserved.

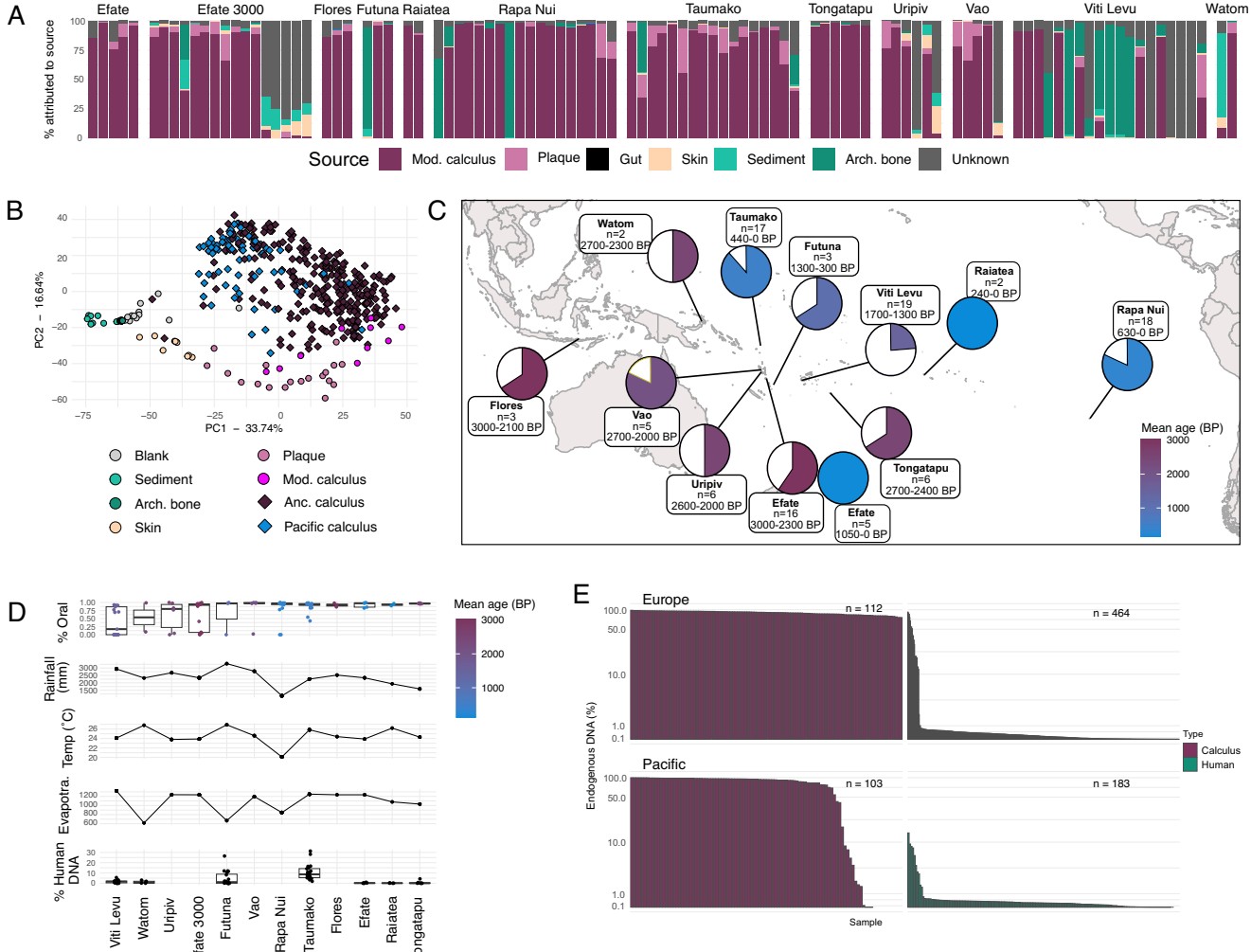

**Fig. 1 | Preservation assessment of dental calculus samples. A** SourceTracker analysis of species tables. Each bar represents a sample, coloured by the proportion of each contributing source (sources are the same as in (**B**)). **B** PCA of well-preserved calculus from the Pacific islands (blue) with samples from the same sources used in SourceTracker analysis plus additional ancient calculus. **C** Map of the islands from which samples were collected for this study, with the island name, number of collected samples, and age of the site. Pie charts indicate the fraction of samples from each site that were considered well-preserved, and are coloured by age of the site. Map was produced using the R packages ggmap, plotly, and sf. **D** Comparison of endogenous oral bacterial DNA and human host DNA in samples with environmental conditions on the islands that may affect DNA preservation. No associations between preservation of either oral bacterial DNA or human host DNA were found with any of the measured environmental factors by ANOVA; values for groups with < 3 samples should be considered unreliable. Boxes show data median, interquartile range (25th–75th percentile) and whiskers indicate minimum and maximum values. **E** Comparison between the Pacific/ISEA and northern Europe (England/the Netherlands) of the percentage of endogenous oral microbiome DNA in ancient dental calculus and human DNA in bones/teeth. Scale is logarithmic. The number of samples in each group is indicated next to the bars. Ancient dental calculus contains high levels of endogenous oral bacterial DNA in the Pacific/ISEA similar to that seen in northern Europe, in contrast to the lower levels of preserved human DNA in bones/teeth from the Pacific islands compared to northern Europe. Arch. bone - archaeological bone; Anc. calculus - ancient calculus; Mod. calculus - modern calculus.

The average GC content of calculus is known to increase with sample age[12,15,16] through the taphonomic loss of AT-rich DNA fragments, and most islands are represented by samples from a single time period, therefore tying island and age/GC content. However, the extent to which loss of AT-rich fragments affects species profiles, perhaps skewing older samples to have higher proportions of high-GC taxa, has not yet been extensively explored. We found that the species with the strongest PC1 loadings (Supplementary Data 3) suggested a taxonomic gradient relating to oxygen tolerance, such as that previously described in calculus samples from the archaeological site of Middenbeemster in the Netherlands[16], but there was no clear association with GC content.

Low taxonomic assignment rates for the Pacific/ISEA samples and the abundance of non-typical oral taxa that predominantly drive separation along PC1 suggest that taxonomic diversity in these samples may not be represented in the genomic databases we used for classification. We performed further taxonomic profiling with additional tools and databases (Kraken2 with two customized RefSeq-based databases, MetaPhlan3), but were unable to resolve these issues, highlighting the difficulty of disentangling undescribed diversity from intrinsic biases in ancient calculus datasets, such as very short average DNA read lengths (≤ 70 bp) (Supplementary Figs. 3, 4).

We performed canonical correlation analysis between PC loadings, environmental metadata, and laboratory characteristics, to determine if preservation may be influencing the calculus community composition (Fig. 2B). Sample loadings on PC1 and PC2 were significantly correlated with average GC content, but not with sample metadata. We further tested how much variance exists in the microbial community between samples all together as well as separated by island, to see whether the heterogeneity of microbial composition,

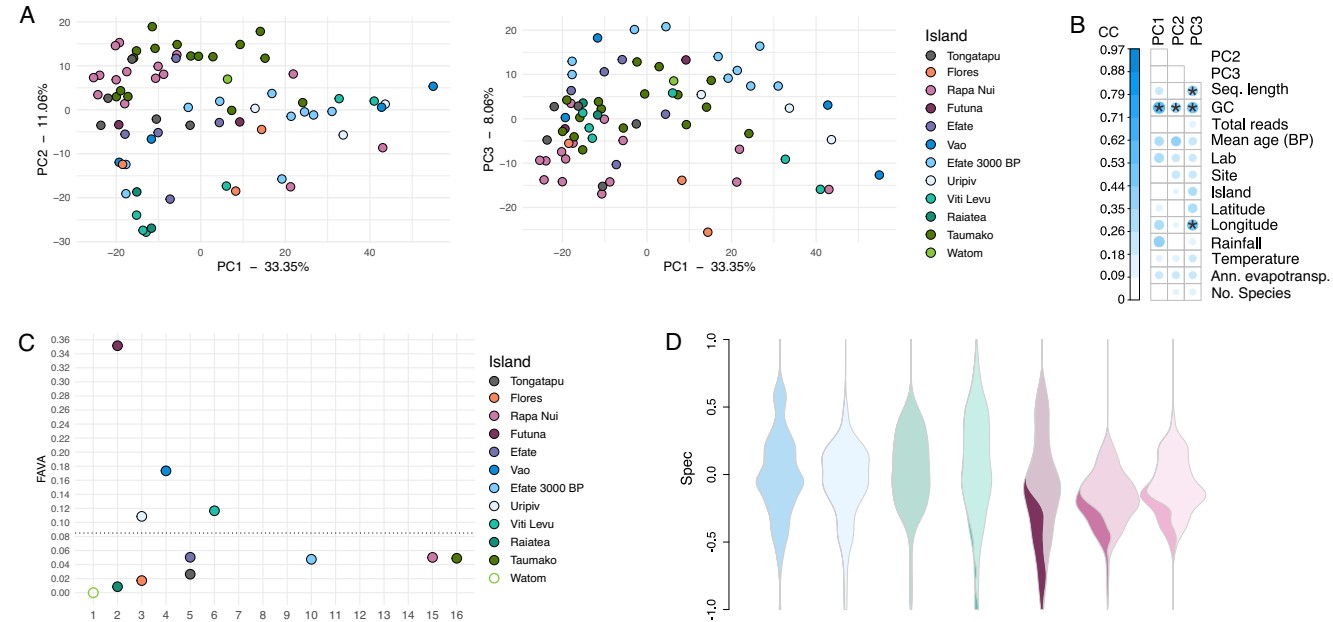

**Fig. 2 | Community species profiles are minimally structured by island. A** PCA of well-preserved calculus samples, coloured by the island from which samples were collected; PCs 1, 2, and 3 are shown, accounting for > 50% of the variation in the dataset. **B** Canonical correlation (CC) analysis comparing the positions of calculus samples in the PCA shown in (**A**) to environmental and laboratory metadata. Pearson two-sided correlation tests were used to determine if the correlations were significant. Metadata with a $p \leq 0.01$ and CC value $\geq 0.4$ are marked with an asterisk (*). Ann. evapotransp. - annual evapotranspiration; No. species - number of species. **C** FAVA values of variance in microbial composition between samples grouped by island. Dotted line indicates the FAVA value of all samples not separated by island. There is only one sample from Watom, so FAVA could not be calculated for this island, and the dot is left unfilled to indicate that FAVA is NA rather than 0. **D** Specificity of species in calculus samples to island conditions (blue), island location (green), or sequenced library characteristics (red). Dark colours in the violin plots indicate the proportion of species that are significantly associated with that metadata type. No dark colour indicates that no species were specifically associated with that metadata.

rather than the dissimilarity measured by PCA, is high. High variability across samples in a group might indicate a community transitioning from one homoeostatic balanced state to another due to a perturbation, potentially indicating a recent community structural shift. Using the R package FAVA[17], we calculated the FST-based Assessment of Variability across vectors of relative Abundances (FAVA) for each island as well as all islands together (Fig. 2C). FAVA values range from 0, indicating identical variance across samples, to 1, indicating maximal variability with each sample having only a single species. The FAVA values are all close to 0, indicating a similar variance in composition within and across islands. However, we note that the islands with smaller numbers of samples have a wider range of FAVA values, while the FAVA value stabilizes above 10 samples, so larger sample numbers for each island are needed to confirm that variability is similar in islands with fewer samples.

We additionally looked for particular species that were significantly associated with environmental and laboratory metadata using the R package specificity (Fig. 2D). No species were significantly associated with rainfall, evapotranspiration, or latitude, while a few were associated with longitude. In contrast, numerous species were significantly associated with sample age, average GC content, and average read length, which are themselves correlated, with some species significantly associated with more than one of these conditions. These results suggest that the environmental conditions tested here have minimal influence on the reconstructed calculus microbiome community, and instead other unexplored factors may be more influential.

### Comparison with global ancient calculus microbiome profiles

As this is the first large ancient dental calculus dataset published from the Pacific/ISEA, we next wanted to know whether the microbiome communities fall within the known variation of published dental calculus datasets from across the globe. We compared the species profiles of the Pacific/ISEA calculus samples to those from Europe, Africa, and Asia using PCA (Fig. 3A). The Pacific/ISEA samples and those from Japan[18] cluster at the same end of the plot, suggesting that this region may have a particular species profile characteristic. Sample clustering based on continent/region (hereafter "continent") was significant by PERMANOVA ($p < 0.01$, F = 2.47, $R^2 = 0.03114$), while tests of beta-dispersion between the continents found differences between Asia and the Pacific/ISEA, Africa, and Europe (Fig. 3B, $p < 0.001$). However, the large difference in sample numbers for each continent make these comparisons less reliable.

The distribution of samples across PC1 in the PCA may be largely driven by the number of species in each sample, as there is a moderate correlation between PC1 value and species counts (Fig. 3C). We confirmed that there is limited variance in the microbial composition of samples within each continent and across all continents by calculating the FAVA for each (Fig. 3D). The values are all close to 0, indicating a similar variance in composition within and across continents, which supports that the calculus microbial community is largely stable across time and geography. The lower average number of species detected in Pacific/ISEA samples compared to other continents may be related to the lower average number of reads in each sample that could be assigned taxonomy (Supplementary Fig. 5), further hinting at unexplored diversity in the Pacific/ISEA calculus.

### Gene content

We next investigated whether microbial gene content distinguishes the Pacific/ISEA calculus samples from those of other continents. Hierarchical clustering of the KEGG orthologs (KOs) detected in samples did not cluster samples by continent or sample age (Fig. 4A), the processing lab, or the study in which they were first presented (Supplementary Figs. 6, 7). Using hierarchical clustering, the samples form

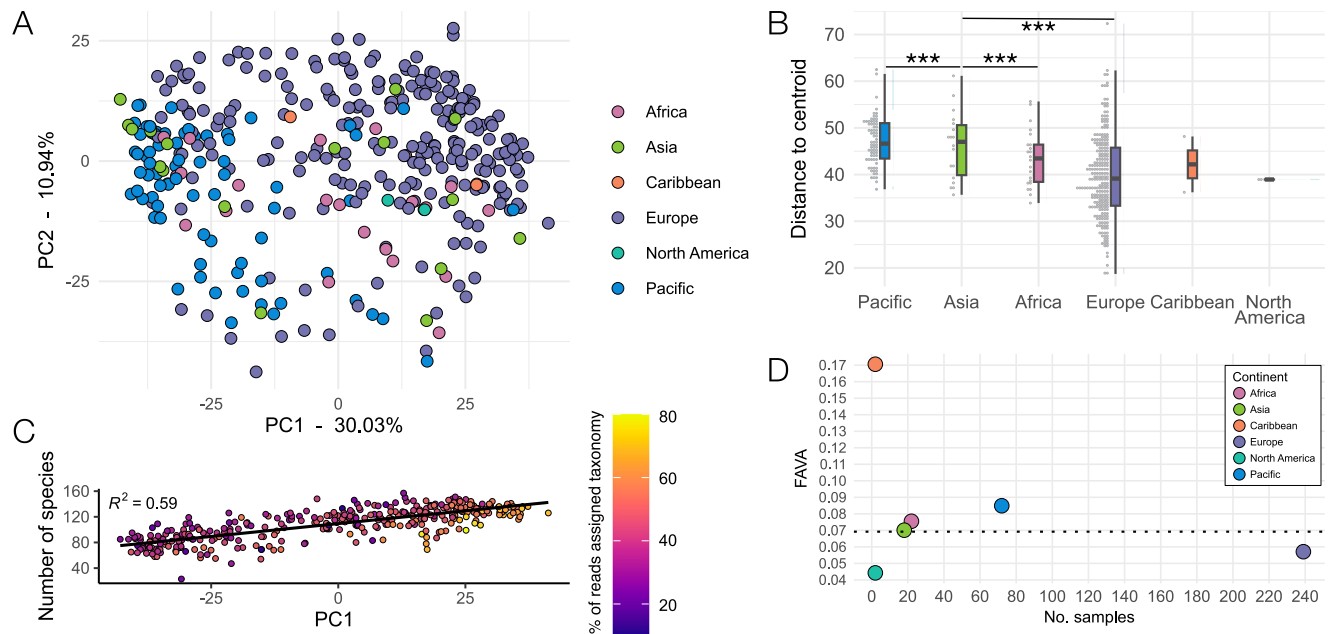

**Fig. 3 | Situating the Pacific/ISEA calculus samples within known ancient dental calculus microbial diversity. A** PCA of Pacific/ISEA calculus samples with ancient dental calculus from additional geographic regions. The additional samples are the same as those in Fig. 1B. **B** The distance to the centroid of all samples in the PCA, test by ANOVA *** *p* = 0.001, with Tukey's Honest Significant Differences; values for groups with < 3 samples should be considered unreliable. Boxes show data median, interquartile range (25th-75th percentile) and whiskers indicate minimum and maximum values. **C** The number of species in each sample, ordered by PC1 loading, and coloured by the percentage of total reads in the sample that were assigned taxonomy. The trend line is fit with a generalized linear model. **D** FAVA values of variance in microbial composition between samples grouped by continent. Dotted line indicates the FAVA value of all samples not separated by continent.

two clusters (Clusters A and B) that correspond to their placement along PC1 in a PCA based on KO abundance (Supplementary Fig. S8A) and based on species abundance (Supplementary Fig. 8B), which is loosely correlated with the number of detectable KOs and species in each sample (Supplementary Fig. S8C). Most of the Pacific/ISEA samples fall in Sample Cluster A, which on average has lower species and KO counts (Supplementary Fig. 8D, E), possibly indicating these samples have more unidentified taxa, with corresponding unannotated genes/metabolic functions.

The KOs formed three clusters (Clusters 1, 2, and 3), and Sample Cluster B was enriched in KOs from Cluster 1 and depleted in KOs from Cluster 3. We grouped the KOs in Clusters 1 and 3 by Pathway and found that one pathway had significantly more KOs in Cluster 1 than 3: Protein families: genetic information processing (Fig. 4B). Given the broadly general cellular processing categories included in this pathway, we assessed the genera that were contributing the orthologs in these pathways to see if we could glean more microbially-relevant information. We found that a high proportion of numerous orthologs were attributed to *Ottowia* (Fig. 4C), specifically *Ottowia* sp. oral taxon 894, a poorly characterized species. The samples in Sample Cluster B have on average higher proportions of *Ottowia* sp. oral taxon 894 than those in Sample Cluster A (Fig. 4C), perhaps indicating a difference in the biofilm environment of these two clusters. However, the presence and abundance of this genus does not appear to drive sample loading in PCA, as the plot structure remains largely unchanged after filtering out *Ottowia* from the input table (Supplementary Fig. 7). Overall, *Ottowia* is the most prevalent genus enriched in KO Cluster 1 that is contributing sample Cluster B (Fig. 4D).

### Phylogenetic analyses

Phylogenetic trees were constructed for *Tannerella forsythia* and *Anaerolineaceae* bacterium oral taxon 439, as both have previously been studied phylogenetically in archaeological dental calculus in relation to human migration[18–21]. In both phylogenetic trees, samples from the same islands generally cluster together (Fig. 5), which was consistent across multiple tree-building methods (Kendall's coefficient of concordance (W) 0.91-0.96 for *Anaerolineaceae* bacterium oral taxon 439 and 0.83-0.98 for *T. forsythia*) (Supplementary Figs. 9, 10). We found little evidence of recombination in either the *T. forsythia* or *Anaerolineaceae* bacterium oral taxon 439 alignments using Gubbins (Supplementary Data 6, 7, Supplementary Fig. 11), and the branches of a ML tree built from the masked SNP alignment was generally concordant with the branches of a ML tree built from an unmasked SNP alignment (Supplementary Figs. 9, 10), but showed greater differences for *Anaerolineaceae* bacterium oral taxon 439.

The percentage of multiallelic SNPs is generally < 20% for *T. forsythia* (Supplementary Data S4), indicating that the reference strain used for mapping is closely related to the strains present in the samples. For *Anaerolineaceae* bacterium oral taxon 439, however, the number of sites with multiallelic SNPs is much higher (Supplementary Data S5), indicating that the reference genome may be quite distinct from the strains present in the samples, and reads from several strains or species in each sample may be aligning to this reference genome. Within a cluster of samples, there is a tendency for samples with higher levels of heterozygosity to fall basal to other samples (e.g., the cluster of samples from Taumako) in both the NJ and ML trees (Fig. 5, Supplementary Figs. 9, 10). It is likely that the *Anaerolineaceae* bacterium oral taxon 439 phylogeny does not represent that of a single taxon, but rather a collection of closely related species or strains, which adds uncertainty to the tree topology. At present, however, only one reference genome is available for oral bacteria in the family *Anaerolineaceae*, making further within-sample taxonomic disambiguation challenging.

We additionally ran inStrain to test the ability to identify shared strains of *Anaerolineaceae* bacterium oral taxon 439 and *T. forsythia* among our samples. Following testing of inStrain with in silico-generated datasets to understand how ancient DNA damage patterns affect the strain assessments (Supplementary Figs. 12, 13), we found identical strains in samples AMH001 and AMH004 (popANI > 99.999). All other samples had popANI values ≤ 99.99, indicating shared, but not identical, strains (Supplementary Fig. 14). The similarity

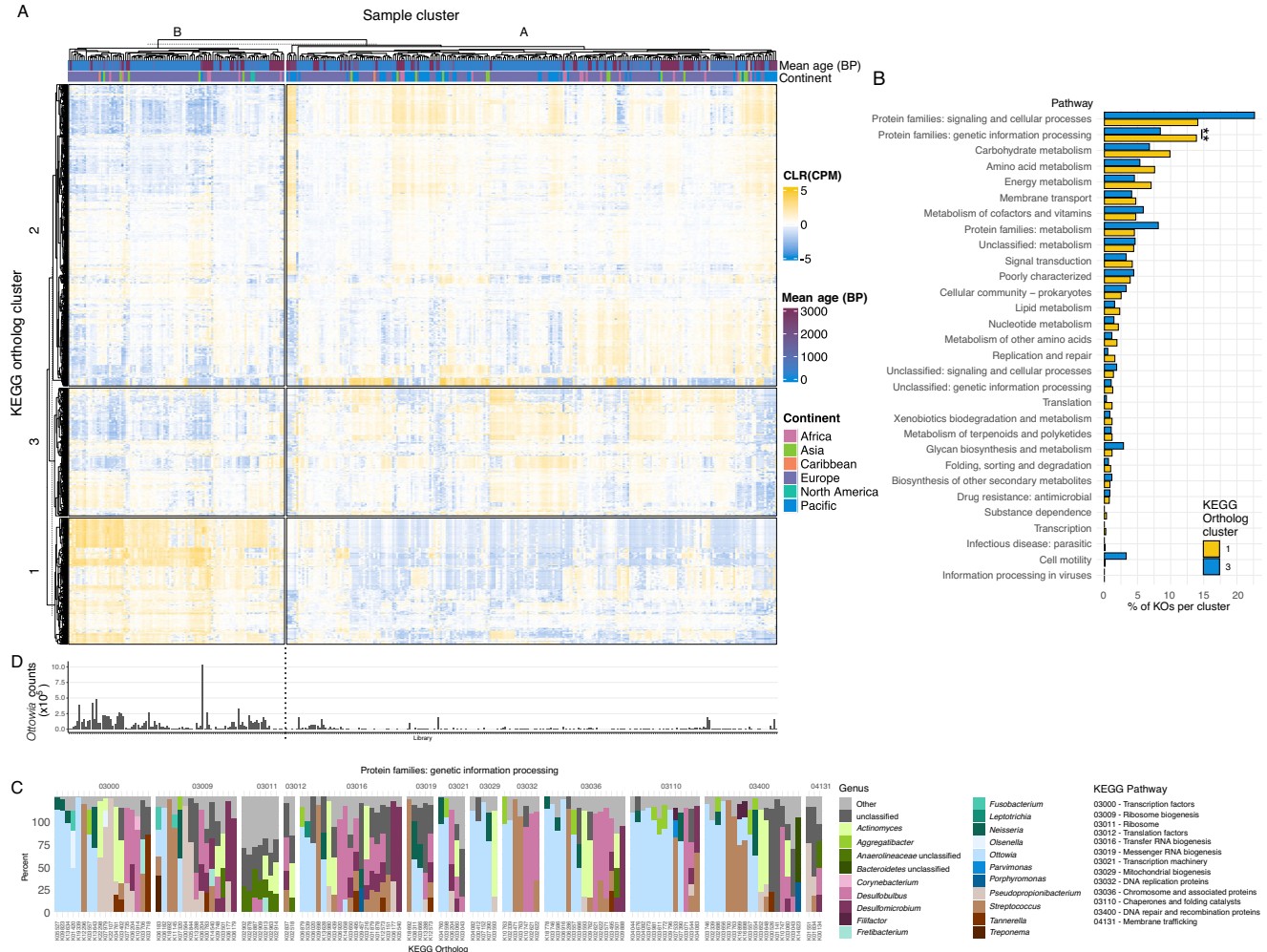

**Fig. 4 | KEGG ortholog (KO) enrichment is associated with sample species composition. A** Heatmap of clustered samples (clusters **A** and **B**) and KOs (clusters 1, 2, and 3), showing CLR-transformed copies per million (CPM) for each ortholog. **B** Percent of KOs in each KO cluster from (**A**) in KEGG pathways present in all samples. Two-sided Wilcox test with FDR correction, ** $p$ = 0.00272. **C** Mean percent contribution by genera of KOs enriched in KO Cluster 1, which are enriched in sample Cluster B. *Ottowia* is the most prevalent genus contributing to these KOs. **D** Read counts of *Ottowia* in all samples, aligned with panel (**A**), showing a higher percentage of samples in cluster B have higher *Ottowia* read counts than in cluster A.

of *Anaerolineaceae* bacterium oral taxon 439 strains (popANI > 99.99%) in samples from the same islands, including Viti Levu (SIG), Taumako (NMU), and Rapa Nui (A*), is reflected in the close phylogenetic clustering of these samples. No identical *T. forsythia* strains were found between any samples (Supplementary Fig. 15), but closely related strains (popANI > 99.91%) were found in samples from Rapa Nui (A*).

As an additional test for whether there are multiple closely-related species or strains of *Anaerolineaceae* bacterium oral taxon 439 in our samples, we calculated the polymorphic rate over protein-coding genes[22] for the reference genomes of *Anaerolineaceae* bacterium oral taxon 439 and *Tannerella forsythia*. The dN/dS values of samples mapped against the *Anaerolineaceae* bacterium oral taxon 439 genome are on average lower than for *T. forsythia*, and fall below the estimated value for mapping to an incorrect reference genome (Supplementary Figs. 16, 17), indicating multiple strains of this species are likely present, while this is less likely the case for *T. forsythia*.

### Dietary DNA
In addition to tracing the microbial changes in dental calculus across the Pacific islands, we sought to examine the potential of recovering eukaryotic, food-derived DNA that may offer insight into dietary patterns across these sites. We were unable to identify any unambiguously positive evidence for dietary DNA in our samples, which may be due to the low number of non-microbial DNA sequences that were recovered. Alternatively, the sequences may be modern contaminants, or they may be aligning to an inaccurate reference genome[23] (Supplementary Data 8).

### Microparticles
To gain further insights into potential dietary patterns in the Pacific, we examined the microparticle content of the dental calculus samples analysed in this study (Supplementary Figs. 18, 19, 20, 21, 22, and Supplementary Data 9). The microparticle results from Rapa Nui[24,25] and Teouma[26] were previously published elsewhere. Overall, the samples had low microparticle counts compared to other dental calculus examined in the Pacific. Almost all samples contained fungal spores and hyphae, likely from sediment. Some starch granules were observed in the samples, but they could not be distinguished from common manufacturing contaminants associated with gloves and laboratory consumables. Phytoliths and diatoms of likely dietary origin were present, but at low levels. This may indicate a greater reliance on starchy root crops and/or processing of plant foods than at Teouma where there were abundant phytoliths recovered from dental calculus[26]. Compared to the published data from Rapa Nui[24,25], where numerous diatoms were found, the data presented here suggests

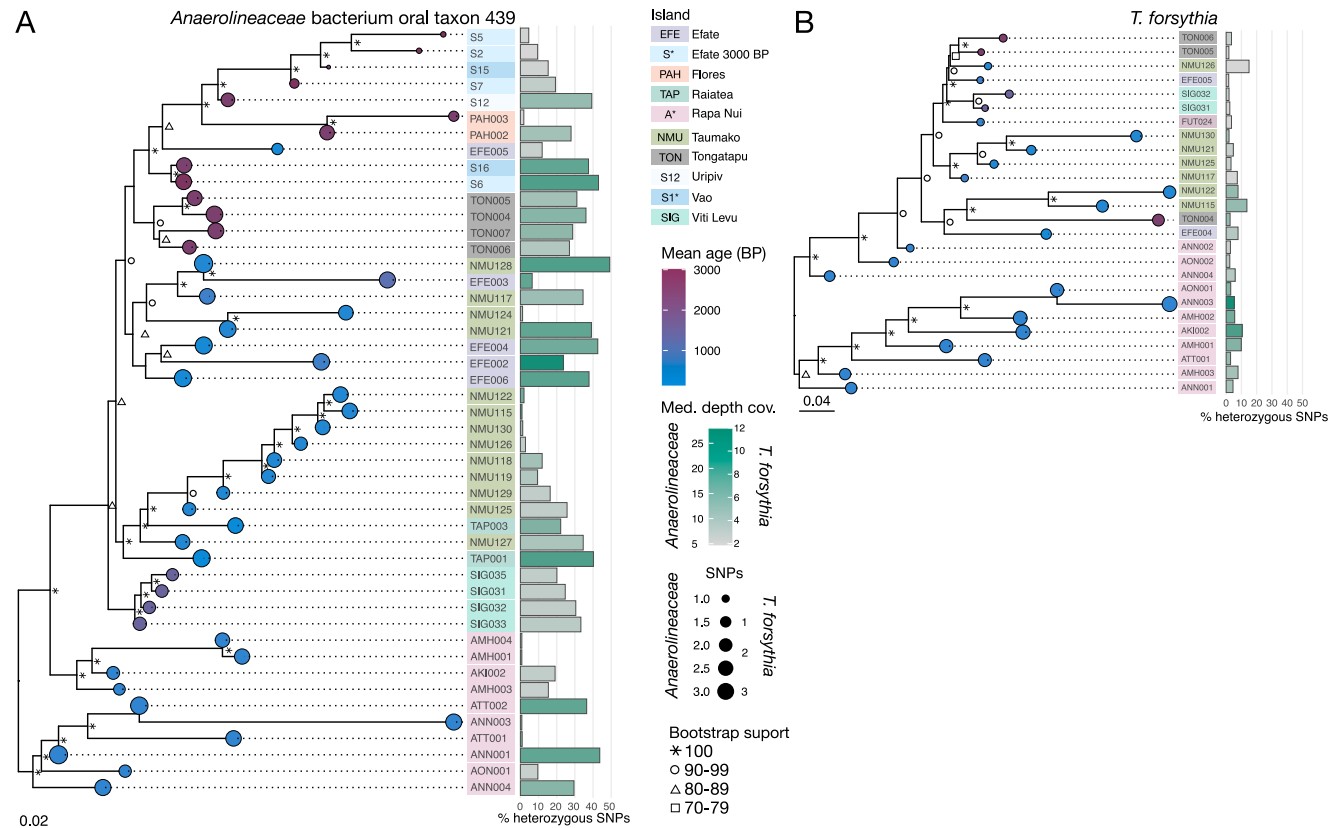

**Fig. 5 | Phylogenetic trees show that bacterial genomes from the same island resemble each other. A** A neighbour-joining tree of *Anaerolineaceae* bacterium oral taxon 439, including only samples with > 5X genomic coverage of the taxon and using only homozygous SNPs, with midpoint rooting. **B** A neighbour-joining tree of *Tannerella forsythia* from samples with > 2X genomic coverage, using only homozygous SNPs with midpoint rooting. For both trees, the age of the sample (in years BP) is shown as coloured circles on tree tips, the circle diameter indicates the number of SNPs (x10,000) in that sample, the island of origin is indicated by a coloured box behind sample IDs, the percentage of heterozygous SNPs is shown as a bar, and the mean coverage of the genome as the colour of the bar. Scale bar indicates the genetic distance.

better freshwater access on these islands than on Rapa Nui. Dietary microparticles were too sparse to draw further conclusions.

## Discussion

Here we show that archaeological dental calculus samples from islands across the Pacific preserve a high proportion of DNA from endogenous oral microbiota, despite a climate that is unfavourable to DNA preservation. This allowed us to assess the diversity of ancient oral microbiomes in an understudied region, to put them in context on a global scale, and to explore the potential of the oral microbiome for tracing human migration through ISEA and the Pacific. While we did not observe temporal or geographic patterns of microbiome species composition in samples from across the Pacific/ISEA or across the globe, we observed a distinct community structure in the Pacific/ISEA calculus compared to samples from Europe spanning a similar time period, suggesting the presence of undescribed microbial diversity in Pacific/ISEA dental calculus oral microbiomes.

Considering the high temperatures and humidity in ISEA and the Pacific, and variable success in human DNA extraction from skeletal elements from the region[1], the overall high preservation of DNA in dental calculus from the Pacific was an unexpected success. These results provide further support to the exceptional preservation of biomolecules in dental calculus[12] compared to bones and teeth. We found that none of the climatic variables we tested predicted preservation, which suggests that smaller-scale local factors, such as soil biogeochemistry, water exposure, or the microclimate of the burial site, may be more influential. Further studies are needed to investigate local factors that may contribute to preservation, which could help

explain why calculus preservation is high even in regions with conditions unfavourable to DNA preservation in skeletal remains.

Given the high level of DNA preservation in ancient dental calculus in the Pacific, we sought to explore the potential of calculus microbial DNA for tracking human migration or other behavioural or cultural changes associated with island colonization. Investigating correlations of host genetic background with community composition or strain sharing, such as by using estimated ancestry proportions for each individual, was not possible in this study, as we did not have enough calculus samples from individuals with paired human genetic data. This is, however, an exciting avenue to explore in future studies. We had limited success identifying differences in the microbial community composition related to island of origin or time period. This is in line with other studies to date, which have not found substantial microbial differences at the community-wide level related to geography, time period, or oral health[11,16,27–29], indicating that the species community composition is relatively stable throughout human history. This is supported by studies of modern oral microbiomes, which indicate that perturbations in the community of dental plaque microbiomes are quickly rebalanced[30] and are generally stable across a variety of cultural practices[31–34], even when gut microbiomes from the same communities under comparison are substantially different.

However, there is growing evidence that the oral species present in ancient dental calculus are not fully represented in current genomic databases, such as NCBI, used for taxonomic profiling. Because of this, community composition analyses may be missing taxa and underestimating diversity, and therefore signals of differences or change may be hidden. The Pacific calculus dataset appears to be particularly

affected by taxonomic database bias, as it has a notably lower taxonomic assignment rate compared to European ancient dental calculus samples for which the distributions of read length and GC content overlap.

Despite the lack of signal at the microbial community level, several studies have demonstrated the utility of phylogenetic reconstruction of abundant species in dental calculus to trace their evolutionary history, including *Anaerolinaeceae* bacterium oral taxon 439 and *Tannerella forsythia* in relation to human migration[11,18–21]. Our own phylogenetic reconstructions of these species did not show clear temporal or latitudinal/longitudinal patterning; however, the reconstructed genomes often clustered by island, suggesting that oral strains are more similar within an island than between islands, which was also supported by independent strain identification with inStrain. In phylogenetic trees for both *T. forsythia* and *Anaerolineaceae* bacterium oral taxon 439, the samples from Rapa Nui, the most remote island in this study, are in two clusters that fall on either side of the midpoint root. This pattern is similar to that of *Anaerolineaceae* bacterium taxon 439 and *T. forsythia* presented by Honap, et al.[21], and suggests that there are 2 distinct lineages of each species present in these samples. Alternatively, these lineages may instead be different species from those of the genomes used as the reference for mapping. However, due to the lack of additional reference genomes for oral *Anaerolineaceae* and *Tannerella*, this possibility is currently difficult to explore.

An outstanding challenge to reconstructing past microbial genomes from metagenomes is distinguishing multiple closely related species and strains within a metagenome. Attempts to study migration patterns through the microbiome, therefore, come with a degree of inherent uncertainty when attributing microbial DNA to particular species and strains. Due to issues such as contaminated ("dirty") reference genomes[35], which include sequences not derived from the species of interest, or incomplete databases[36], it is possible that sequencing reads from multiple species are inappropriately aligned to a reference genome[23,35], creating noise in the data analysis. Therefore, the reliability of conclusions regarding human migration drawn from trees built using our current approaches is uncertain.

This issue appears to have particularly affected our reconstruction of *Anaerolinaeceae* bacterium oral taxon 439, for which we observed high rates of SNP heterozygosity. As there is only a single isolate reference genome of *Anaerolinaeceae* bacterium oral taxon 439 sequenced to date, the extent of diversity in this organism both past and present is yet unknown. The high rate of SNP heterozygosity in many samples mapped to this reference genome appeared to affect the branching pattern within clades, despite inclusion of only biallelic SNPs in the alignment used to build our tree. Future sequencing of additional isolates of this species, or reconstruction of metagenome-assembled genomes (MAGs) of this organism from deeply sequenced modern and ancient oral metagenomes[37], may lead to more accurate strain separation, read alignment, SNP calling, and phylogenetic tree reconstruction. As the study of migratory patterns through ancient host-associated microbiomes is still in its infancy, method development will be fundamental in order to explore the full potential of this field.

Our results indicate the high potential of dental calculus to be well-preserved in geographic and climatic conditions that are otherwise unfavorable to DNA preservation, opening the possibility to explore archaeogenetics data in formerly poorly-accessible locations. Although we did not observe the microbial community composition of the calculus microbiome structuring by island or time period, the low taxonomic assignment rate suggests that there is additional taxonomic diversity in the Pacific calculus samples beyond that currently represented in databases, highlighting the need for studies of dental plaque biodiversity in broad, global contexts. Individual species reconstructions have the potential to reveal evolutionary patterns that mirror the migration patterns of their human hosts, but further work

disentangling closely related species and strains within ancient dental calculus metagenomes, as well as revealing currently unknown species diversity, is needed to allow accurate identification of individual species and to perform reliable phylogenetic reconstructions.

## Methods

### Ethics & inclusion statement

Archaeological research was carried out in close consultation with local communities and in partnership with local cultural councils, museums, and research institutions. Local researchers contributed to the study as co-authors. Export of samples for analysis was approved as part of the permissions processes described below.

**Efate, Vanuatu.** The Vanuatu Cultural Centre (VCC) and the Vanuatu National Cultural Council provided ethical oversight of the study of human remains from Efate. In addition, the leaseholder M.R. Monvoisin and family and the traditional landowners and population of Eratap Village provided support for the study of remains on their ancestral lands. Monica Tromp entered into a research agreement in 2013 with the Vanuatu Cultural Centre to conduct research on human dental calculus from multiple archaeological sites in Vanuatu. The permit number is 10071.

**Flores, Lesser Sunda Islands, Indonesia.** The Pain Haka cemetery on the island of Flores in Indonesia was discovered and excavated by archaeologist Jean-Christophe Galipaud and anthropologists Charles Illouz and Philippe Grangé in 2010 (Institute for Research for Development [IRD] and University of La Rochelle) under a research permit from the Province of Nusa Tengarra Timur. Further excavation and sampling in 2012 were made possible by a joint collaboration between IRD, Puslit Arkenas, Hallie R. Buckley, and University of Otago. Export permits were obtained from Puslit Arkenas and RISTEK (Foreign Research Permit Division, Ministry of Research and Technology/National Research and Innovation Agency). Local communities participated at all stages of the research and gave permission for the removal of samples for analysis. All samples were collected and prepared by Rebecca Kinaston and exported to Otago University for management.

**Futuna, Vanuatu.** The Vanuatu Cultural Centre (VCC) and the Vanuatu National Cultural Council provided ethical oversight of the study of human remains from Futuna. Monica Tromp entered into a research agreement in 2013 with the Vanuatu Cultural Centre to conduct research on human dental calculus from multiple archaeological sites in Vanuatu. The permit number is 10071.

**Raiatea, Society Islands.** Excavations at Taputapuatea on the island of Raiatea were undertaken in 1994 and 1995 by archaeologists of the Centre Polynésien des Sciences Humaines (Tahiti, Polynésie française), and the export permit for subsequent analysis was issued by Service de la Culture et du Patrimoine (Tahiti, Polynésie française) in 2014. More recent communication (March 2020) with Director of Direction de la Culture et du Patrimoine (Tahiti, Polynésie française), granted permission for human dental calculus analyses.

**Rapa Nui, Chile.** Archaeological remains were sampled in 2002 as part of John Dudgeon's (2008) dissertation research from collections excavated during the National Geographic Easter Island Anthropological Expedition. Led by George Gill of the University of Wyoming, Sergio Rapu, former curator of the Sebastian Englert Museum, and Claudio Cristino of the University of Chile, the expedition collected the skeletal material in several field seasons, ending in 1981. The skeletal remains were curated at the Museo Antropológico Padre Sebastián Englert (MAPSE), Rapa Nui. Approval for the collection of skeletal remains for analysis was approved by the Consejo de Monumentos Nacionales de Chile, and the Museo Antropológico Padre Sebastián

Englert, under then-director Francisco Torres Hochstetter. Skeletal materials from which dental calculus was extracted were repatriated to Rapa Nui in 2009.

**Taumako, Duff Islands, Solomon Islands.** The skeletal remains from Taumako in the Solomon Islands were excavated in the early 1970s by Foss Leach and Janet Davidson. We do not have any record of the original research permit. The skeletal remains were curated at the Anatomy Dept in Otago University firstly by Professor Phil Houghton and then by Hallie Buckley. The skeletal remains were repatriated in 2009 to the Solomon Islands National Museum. At that time permission was given to retain bone and tooth samples for further destructive analyses by the Director Lawrence Kiko.

**Tongatapu, Tonga.** Excavations at Talasiu (Tongatapu) were directed by archaeologist Geoffrey Clark and human and cultural remains export were permitted by Tongan Traditions Committee (Komoti Talafakafonua), Nuku'alofa, Kingdom of Tonga (2016-2025). The excavations of human skeletal remains at Talasiu were conducted in consultation and with the agreement of the Lapaha community.

**Uripiv, Vanuatu.** The Vanuatu Cultural Centre (VCC) and the Vanuatu National Cultural Council provided ethical oversight of the study of human remains from Uripiv. Monica Tromp entered into a research agreement in 2013 with the Vanuatu Cultural Centre to conduct research on human dental calculus from multiple archaeological sites in Vanuatu. The permit number is 10071.

**Vao, Vanuatu.** The Vanuatu Cultural Centre (VCC) and the Vanuatu National Cultural Council provided ethical oversight of the study of human remains from Vao. Monica Tromp entered into a research agreement in 2013 with the Vanuatu Cultural Centre to conduct research on human dental calculus from multiple archaeological sites in Vanuatu. The permit number is 10071.

**Viti Levu, Fiji.** The skeletal remains from the Sigatoka Dunes site on the island of Viti Levu in Fiji were sampled in 2015 from prior excavations curated at the Fiji Museum, Suva, Fiji. The original excavations were conducted under the Sigatoka Salvage Archaeological Project by Simon Best in 1987 and 1988. Sampling of skeletal remains was approved by the Fiji Ministry of Education and the Immigration Department and by the Fiji Museum, and sampling was assisted by museum staff Sepeti Matararaba, Jone Balenaivalu, Elia Nakoro, Sakiusa Kataiwi, and Jotami Naqeletia. Funding for this research was provided by the Nation Science Foundation of the United States, Award # SBS 1216310.

**Watom, Bismarck Archipelago, Papua New Guinea.** Excavations at Watom in 2008 and 2009 were directed by archaeologist Dimitri Anson and Hallie R. Buckley. Research permits for the excavation included permissions for export of human and cultural remains for analysis. In 2008, Hallie Buckley obtained permission from Herman Mandui, Chief Archaeologist of the National Museum and Art Gallery of Papua New Guinea in 2008. We have also had more recent communications (2023) with the Director of the Museum, Alous Kuaso, granting permission for further DNA and other destructive analyses. The excavations of human skeletal remains on Watom Island were conducted in consultation and with the agreement of the Village community.

## Laboratory methods

A total of 102 archaeological dental calculus samples were processed in this study (Table 1, Supplementary Data 1). The dental calculus samples were processed in three groups, hereafter referred to as Jena, Oklahoma and Otago, named after the respective processing lab. Each sample group was processed using a slightly different extraction and library preparation protocol. Samples from Efate were analysed in two groups - Efate (samples processed in Jena, <1050 BP) and Efate 3000 BP (samples processed in Otago, 3000-2300 BP). Temporal information for the dental calculus samples in this study were obtained either through direct radiocarbon dating of the individual or by cultural association of the burial. Sample extraction and library preparation followed Dabney, et al.[38] with slight variations between labs.

**Jena.** All sample processing took place in a dedicated cleanroom facility at the Max Planck Institute for Evolutionary Anthropology (MPI-EVA) laboratories (Jena, Germany). Total DNA was extracted from 0.5-7 mg of dental calculus per individual, using a silica column-based extraction protocol optimized for the recovery of short DNA fragments, adapted for dental calculus following the methods described in refs. 12,38,39 and available as downloadable bench protocol through the online protocol repository protocols.io at https://doi.org/10.17504/protocols.io.bidyka7w[39]. Blanks were processed alongside the samples. The extracts were prepared into double-stranded libraries with partial uracil-DNA-glycosylase (UDG) treatment[40] and dual indexing[41,42] following protocols available through protocols.io at https://doi.org/10.17504/protocols.io.bmh6k39e and https://doi.org/10.17504/protocols.io.4r3l287x3l1y/v3[43,44]. The libraries were sequenced to a depth of $10.5 \pm 2.3$ million reads (mean ± standard deviation) on an Illumina Nextseq with 75-bp paired-end sequencing chemistry. Blanks were processed alongside the samples for both extraction and library preparation. Two samples, SIG040 and SIG046, were extracted and sequenced twice, as it was suspected that burial sand may have been unintentionally included during the first processing round.

**Oklahoma.** Total DNA was extracted from 0.8–12.8 mg dental calculus per individual at the ancient DNA facility of the University of Oklahoma Laboratories of Molecular Anthropology and Microbiome Research (LMAMR, Norman, OK, USA), using a protocol very similar to that used in Jena. To remove surface contaminants, dental calculus samples were UV-irradiated for 1 min on each side, and washed with 1 ml 0.5 M EDTA for 15 min. The decontaminated calculus was then resuspended in 1 ml of 0.5 M EDTA solution and incubated overnight at room temperature. A 100 µl proteinase K solution (> 600 mAU ml−1; Qiagen, Cat. No. RP103B) was added and incubated at 37 °C for 8 h, followed by continued digestion under agitation at room temperature until decalcification was complete. After digestion, the supernatant was added to 14 mL of PB buffer (Qiagen, Cat. No. 19066). This was then centrifuged in a MinElute column (Qiagen, Cat. No. 28006) attached to a Zymo-Spin V column (Zymo Research, Cat. No. C1012) for 4 min at 1500 rcf, rotated, and then centrifuged for an additional 2 min. The column was then dry spun for 1 min at 3400 rcf and washed twice with 700 µL PE buffer (Qiagen, Cat. No. 19065) at 9400 rcf. DNA was eluted from the column after a 5 min RT incubation into 30 µL of EB buffer (Qiagen, Cat. No. 19086) under centrifugation at 17900 rcf. Blanks were processed alongside the samples. The extracts were thereafter shipped to MPI-EVA, where they were prepared into libraries, as described above, alongside the Jena samples. The samples were sequenced to a depth of $10.6 \pm 1.5$ million reads on the same NextSeq flow cells as the Jena samples with 75-bp paired-end sequencing chemistry.

**Otago.** Total DNA was extracted from approximately 1–17 mg of dental calculus per individual using a phenol-chloroform aDNA extraction protocol[45]. In brief, dental calculus samples were washed with ultrapure water and allowed to dry in a laminar flow hood overnight. A second wash was performed using 1 ml of 0.5 M EDTA, with a 30 min incubation time. The supernatant was removed, and the samples were demineralized in 1 ml of 0.5 M EDTA for up to 72 h, until fully demineralized. The supernatant was added to a tube with 750 µl of

phenol:chloroform:isoamyl (25:24:1), vortexed, and left on a rotator for 10 min. After centrifuging, the aqueous phase was transferred to 750 μl of phenol:chloroform:isoamyl alcohol (25:24:1). The incubation step was repeated, after which the aqueous phase was transferred to 750 μl of chloroform:isoamyl alcohol (24:1). After vortexing and mixing by inversion, the mixture was centrifuged and the aqueous phase transferred to 13 ml of 6 M GuSCN and 200 μl of silica suspension, and left on a nutator for 30 min. After centrifugation, the supernatant was removed, the silica was resuspended in 1 ml of GuSCN binding buffer, and the supernatant discarded after centrifugation (three times in total). The silica pellet was air dried for 15 min, and DNA eluted twice in 60 μl TE (heated to 65–75 °C). Blanks were processed alongside samples through extraction and library preparation. Double-stranded libraries were prepared by blunt-end repairing the DNA strands, and after that ligating and filling in adaptors. The libraries were amplified using KAPA HiFi DNA polymerase (Roche, 07958927001), and no UDG-treatment was performed. The libraries were sequenced using an Illumina MiSeq 75-bp paired-end sequencing chemistry to $8.3 \pm 7.2$ million reads at the Otago Genomics Facility (Otago, New Zealand).

### General data processing
Data analyses were conducted in R v.4.1.0[46], unless otherwise stated. General packages used were *tidyverse* v.1.3.1[47], *readxl* v.1.3.1[48], *ggpubr* v.0.4.0[49] and *janitor* v.2.1.0[50]. The colour palette for the study is from the R package *microshades* v.0.0.0.9000[51]. Regression models were drawn to data with a generalized linear model with geom_smooth in ggplot2 as part of tidyverse.

### Preprocessing
DNA sequencing data was preprocessed using the nf-core/eager v.2.3.3 pipeline[52]. Default options were used unless otherwise stated. Poly-G stretches were removed from the raw data, as they are a common by-product of the two-color chemistry sequencing strategy used by Illumina's NextSeq. Human DNA was removed from the dataset by mapping to the human reference genome GRCh38, and only unmapped reads were retained for downstream microbiome analyses. Taxonomic profiling to produce an OTU table was performed using MALT v.0.4.1[53,54] with a custom database[11]. The database contained all bacterial and archaeal assemblies (scaffold/chromosome/complete levels, up to 10 randomly selected genomes per species) from RefSeq and the human HG19 reference genome via the nf-core/eager pipeline. In addition, the dataset was aligned to the NCBI nt database (as of October 2017), to screen for eukaryotic DNA. MEGAN v.6.17.0[55] was used to export OTU tables from the resulting MALT-produced rma6 files, using summarized read counts at both the genus and species level.

Comparative datasets of published microbiome studies were also processed using the same procedures. One comparative dataset was used to assess preservation with the programme SourceTracker[14], and consisted of 10 non-industrialized gut samples[56,57], 10 industrialized gut samples[58,59], 10 skin samples[60], 10 subgingival and 10 supragingival plaque samples[58], 10 femur archaeological bone samples[11], 10 modern dental calculus samples[11] and 10 archaeological sediment samples[61] (Supplementary Data 2). In addition, 10 archaeological petrous bone samples from Taumako and 10 from Viti Levu were included as local environmental controls; this data had been produced during genetic screening of human remains for human population genetic studies at MPI-EVA laboratories in Jena, Germany. Because bones are free of microbes during life, microbes detected in these samples provide a good proxy for local post-mortem colonization[62].

A second comparative dataset was used to compare calculus species profiles across the globe, and consisted of ancient calculus samples from 6 previously published studies. These included samples from Japan[18], Europe and Africa[11], Europe[16,20,27], and Europe, the Caribbean, North America, and Mongolia[12] (Supplementary Data 2).

### Preservation
Preservation was assessed using SourceTracker v.1.01[14], PCA, and the R package cuperdec[11,63] by comparing the microbial DNA in this study to previously published metagenomic datasets[1,5,16,27,64,65]. For Source-Tracker analysis, a species-level OTU table was used as input, and the published reference metagenomic datasets were used as sources. Rarefaction was performed to 10,000 reads, with a training data rarefaction of 5,000 reads. For principal component analysis (PCA), species-level read counts of all dental calculus samples and sources (including 9 modern dental calculus samples) were compared. The R-package cuperdec[11,63] was used to identify well-preserved samples (adaptive burn-in method, cut-off 50%), and 73 (out of 102) samples were carried on to further analyses based on this analysis.

Putative environmental and laboratory contaminants in the dental calculus samples were identified using the R package *decontam* v.1.6.0[66], with the prevalence method. The samples were separated into groups based on the processing lab, and blanks and archaeological bones were used as proxies for contamination sources (cut-off 0.25 for all groups, for both blanks and bones). Contaminants from all labs were combined into a single list of all contaminants, and all contaminants were removed from all samples. The comparative calculus datasets were likewise assessed with decontam and all contaminants were combined into a single list, along with those from the Pacific calculus dataset here, and all contaminants were removed from all datasets for the comparative analyses.

To evaluate whether preservation was related to environmental conditions, a dataset consisting of annual average temperature and annual total rainfall was compiled for each island in this study[67]. Missing data for some islands was obtained from alternative sources[68,69]. Annual average evapotranspiration was compiled from[70].

The proportion of taxa estimated to originate from the oral microbiome (i.e., dental plaque and dental calculus) by SourceTracker was used as a proxy for the preservation of the archaeological dental calculus samples. The effects of environmental variables and/or sample age on preservation were investigated using beta regression with a complementary log-log link function to account for observed heteroscedasticity. Using ANOVA, it was found that the model was not significantly improved by adding the random effects of laboratory and/or island (p»1). Step AIC (using both directions) was used for model selection. To reliably estimate parameters for the model, statistically influential data points were removed, and the model-fitting process was repeated until a stable dataset was reached. Model fit was measured using the $R^2$ value as suggested by Ferrari and Cribari-Neto[71] for beta regression models.

Preservation of calculus was determined by the percentage of the sample that was determined to be from an oral source (calculus, supragingival plaque, and subgingival plaque) with SourceTracker. Preservation of human DNA was determined by the percentage of endogenous human DNA in shotgun sequenced data, and was taken from published values. Human DNA preservation in the Pacific was taken from Posth, et al. (Supplementary Data 3)[1] and Liu, et al. (Supplementary Data 2)[5], human preservation in England was from Schiffels, et al. (Supplementary Data 1)[64] and Patterson, et al. (Supplementary Data 1)[65], and calculus preservation in Europe was taken from SourceTracker values of data from[16,27].

### Taxonomic profiling
Taxonomic profiles were generated with three tools, MALT, Kraken2, and MetaPhlAn3. The MALT table was generated as described above as part of the nf-core/eager run, and was used for all community composition analyses. The tools Kraken2 and MetaPhlAn3 were used to assess whether altering parameters or using non-standard databases increased the number of taxonomic assignments in samples (Supplementary Fig. 3, Supplementary Fig. 4). Kraken2 was run with default parameters, using two databases (described in ref. 34): a custom RefSeq database and the same custom RefSeq database with the

addition of MAGs from[22]. This allowed us to test whether including additional diversity in the database resulted in a substantial increase in the number of reads assigned to taxonomy. However, we found that it did not, and that this profiler is particularly affected by the read length. MetaPhlAn3 was run with two different sets of parameters: default, and custom settings optimized for assignment of ancient DNA (-D 20 -R 3 -N 1 -L 20 -i S,1,0.50 and minimum read length of 35 bp). The use of ancient-optimized parameters substantially increased the number of species identified in each sample (Supplementary Fig. 4A) and we found that the number of species identified by MetaPhlAn3 with ancient-optimized parameters and by MALT were correlated (Supplementary Fig. 4B). The optimal number of sample clusters within the Pacific calculus dataset was determined using the Gap statistic with clusGap from the R package cluster[72] with 500 bootstrap replicates, on both the same MALT species table used for compositional analysis, as well as on a MALT genus table that was cleaned and filtered following the same steps as the species table. The optimal number of clusters was found to be one for both the species and genus tables (Supplementary Fig. 2), so no further cluster analysis was performed.

## Community composition

Principal component analysis (PCA, Euclidean distance) was conducted on the decontaminated species table of only the Pacific data, and again on a decontaminated species table of the Pacific data plus the global comparative data, using the R package MixOmics (Supplementary Data 3) with a centred-log ratio transformation. Drivers of variation in the community composition were tested with a PERMANOVA from the R-package vegan v.2.5.7[73], with euclidean distances and 999 permutations. Homogeneity of multivariate dispersions was tested with the function betadisper from the R package vegan. Canonical correlation was performed using the R package variancePartition[74]. To determine if particular species were associated with environmental conditions on the islands, geographic location, sample age, or library characteristics, we used the R package specificity[75] and the decontaminated species table. We used the R package FAVA[17] to test variability between samples both within and across islands for the Pacific dataset, and within and across continents for the global dataset. Pacific calculus sample clustering was performed following Quagliariello, et al.,[29] with optimal sample cluster number determined using the function clusGap in the R package cluster[72], however the optimum number of clusters was determined to be 1 and no further cluster analysis was performed (Supplementary Fig. 2). For details see Supplementary Methods.

## Functional analysis

Functional analysis was performed using HUMAnN3[76]. All reads < 50 bp were removed from the fastq files prior to running HUMAnN3, because these are generally too short to be classified after translation. The bowtie2 mapping parameters were adjusted to account for aDNA damage patterns (-D 20 -R 3 -N 1 -L 20 -i S,1,0.50). The standard gene family output table with UniRef90 gene clusters was grouped to KEGG Orthology, and analysis was performed on the KEGG orthologs. Poorly preserved samples as well as those with < 750 orthologs were removed, and orthologs present at < 0.005% abundance were filtered out prior to analysis. The optimal number of sample clusters and KEGG Ortholog clusters was determined using the eclust function of the R package factoextra[77] with kmeans clustering, which uses clusGap of the cluster[72] R package. The gap statistic was visualized with the factoextra function fviz_gap_stat, while clustering of sample and KEGG Orthologs was visualized from the output of the eclust function using ggplot. For sample clustering, the optimal cluster number was determined to be 6 (Supplementary Fig. 6A), however, visualization of the clustering demonstrated substantial overlap of the samples of each cluster (Supplementary Fig. 6B). Based on this result, for hierarchical clustering, a sample cluster number of two was chosen to reflect the trajectory of samples in Supplementary Fig. 6B. For KEGG Ortholog

clustering, the algorithm did not converge after 10 iterations, the maximum number of clusters tested. Visualization of the ortholog clustering with 10 clusters (Supplementary Fig. 6C) revealed substantial overlap of the orthologs in different clusters, yet one cluster (cluster 5) included orthologs that clearly separated into two clusters (Supplementary Fig. 6D). Based on visual inspection of Supplementary Fig. 6D, three was selected as the number of ortholog clusters to include in hierarchical analysis.

A PCA was performed using the same steps as with taxonomy. PERMANOVA was performed with the adonis function in the R package vegan[73]. Hierarchical cluster analysis was performed within the Heatmap function of the R package ComplexHeatmap[78]. Both samples and KEGG orthologs were clustered with the "complete" method, and the number of clusters was specified based on the clustering and Gap statistic testing done in the preceding paragraph (2 clusters for samples, and 3 clusters for orthologs).

## Phylogenetic analyses

The nf-core/eager pipeline was used, as described above, to map the non-human reads of the well-preserved samples to the abundant and prevalent oral bacteria *Tannerella forsythia* (strain 92A2, assembly GCA_00238215.1) and *Anaerolineaceae* bacterium oral taxon 439 (assembly GCA_001717545.1). Through nf-core/eager, duplicates were removed using Picard MarkDuplicates v.2.22.9, and prior to mapping, damage was clipped off of the reads (two bases for libraries with partial UDG treatment, and seven bases for non-treated libraries). Genotyping was performed with GATK UnifiedGenotyper, allowing for heterozygous calls and using all sites, with the SNP likelihood model. A minimum base coverage of 5 was required. The SNPs were further filtered in order to construct the phylogenies with only homozygous SNPs (defined as the major nucleotide having a frequency greater than 0.9), using MultiVCFanalyzer v.0.0.87[79] (Supplementary Data 4, 5). Only samples with at least 1000 SNPs and a mean genome-wide coverage of at least 2X (for *T. forsythia*) or 5X (*Anaerolineaceae* bacterium oral taxon 439) were included. The coverage requirement was increased for *Anaerolineaceae* bacterium oral taxon 439, because its percentage of heterozygous SNPs was higher.

Recombination was assessed using Gubbins[80] v3.3.5 on a full alignment of *Anaerolineaceae* bacterium oral taxon 439 and *T. forsythia*. Within Gubbins the initial tree was built with FastTree[81] and subsequent trees were built with RaxML[82] v8.2.12, using the best model, GTRGAMMA, as determined by iqtree[83] v2.3.4 and pyjar[84] v1.0, and 100 bootstrap replicates. RaxML was independently run in RaxML-NG[85] v1.2.2 to build trees with 200 bootstrap replicates on the same SNP alignment that was used to build the neighbour-joining tree, as well as on the masked alignment produced by Gubbins. Neighbour-joining trees and maximum likelihood trees were built using distance matrices generated with the TN93 + G4 substitution model, as this was determined by testing with the DECIPHER R package[86,87] to be the best model for each dataset, with 200 bootstrap replicates. The trees were rooted using the midpoint, determined with the R packag phangorn[88]. The neighbour-joining phylogenetic trees were constructed and visualized using R packages ape v.5.5[89], ade4 v.1.7.17[90], adegenet v.2.1.3[91] and ggtree v.1.99.1[92], while maximum likelihood trees were built with RaxML-NG[85] v1.2.2 using RaxML v8[82]. Tree concordance by Kendall's coefficient of concordance (W) with the function CADM.global and by a Mantel test with CADM.post in the R package ape[93]. For this, trees were converted to distance matrices with the ape function cophenetic, and distance matrices of the trees being compared were joined as a single file with rbind. The combined distance matrices were used as input for both CADM.global and CADM.post with 99 permutations.

## Strain sharing

Strain sharing across samples was assessed with inStrain[94]. A test dataset was generated in silico to test for the effects of ancient DNA

damage on popANI calculations. For this, two 10 M bp paired-end simulated read datasets were generated to test the effects of aDNA damage patterns and clipping, using the genomes of the 10 species in the ZymoBiomics kit that was used by Olm, et al. for testing, in the proportions described in the kit. The genomes were processed with gargammel[95] in two batches, to create datasets with two different damage profiles based on the damage of the samples sequenced here. Two samples were selected for references, HCLVMBCX2-3505-07-00-01_S8, which is non-UDG treated, and EFE002.B0101, which is UDG-half treated. Both samples were mapped against the *Anaerolineaceae* bacterium oral taxon 439 genome (GCA_001717545.1) with bwa (-n 0.02 -l 1024)[96], and the bam files were used as input to mapDamage[97] to generate damage profiles and read length distributions that were used as input to gargammel. The simulated datasets were processed with the nf-core/eager pipeline the same way as samples, and adaptor-trimmed un-collapsed paired reads were pulled out of the eager output with a custom script. The paired end reads were mapped against all 10 reference genomes in the ZymoBiomics kit, combined into a single fasta file with bwa (-n 0.02 -l 1024). The effect of read end masking to remove aDNA damage was tested with a custom python script[98], with mapped reads from UDG-treated reads masked 1 base on either end, and non-UDG-treated reads masked 9, 11, 13, and 15 bases on either end, based on the C-T transversion ratio along read lengths determined by mapDamage. InStrain profile was run on the unmasked and masked mapped read bam files with insert sizes 12, 24, 36, and 48, and inStrain compare was run on all output of the profile step. Based on the output of these simulated data tests, a mask length of 11 and insert size of 12 were selected for processing real samples, as these returned a maximum popANI with minimal loss of coverage.

The coverage overlap and popANI between all samples was visualized and compared in R (Supplementary Figs 12 and 13).

Strain sharing across Pacific calculus samples was assessed with inStrain for two species that were highly abundant across the Pacific calculus dataset: *Anaerolineaceae* bacterium oral taxon 439 (CP017039.1), and *Tannerella forsythia* (NC_016610.1). Paired-end reads were mapped against reference genomes for *Actinomyces dentalis* (GCF_000429225.1), *Anaerolineaceae* bacterium oral taxon 439 (CP017039.1), *Desulfobulbus oralis* (CP021255.1), *Eubacterium minutum* (CP016202.1), *Olsenella* oral taxon 807 (CP012069.2), and *Tannerella forsythia* (NC_016610.1), combined into a single fasta file. Only *Anaerolineaceae* bacterium oral taxon 439 and *Tannerella forsythia* had sufficient genome breadth and depth of coverage across enough samples for reliable strain comparison with inStrain. Mapped reads in each bam file were masked according to their UDG treatment, with UDG-half-treated samples masked 1 base at both ends, and non-UDG-treated samples masked 11 bases at both ends. inStrain profile was run on the masked bam files with an insert size of 12, and the default value of 5X minimum coverage was required for samples to be included in inStrain analysis. inStrain compare was run on all output files of inStrain profile. We focused only on *Anaerolineaceae* bacterium oral taxon 439 and *Tannerella forsythia*, for which we generated whole genome SNP-based phylogenies. The script polymut.py from cmseq[22] was used to assess whether samples contained multiple strains of the species *Tannerella forsythia* and *Anaerolineaceae* bacterium oral taxon 439 based on the ratio of non-synonymous vs. synonymous sites (dN/dS) in coding regions of the genome. This is a previously published ad-hoc method[22] to assess strain heterogeneity in genomes produced from metagenomes. Testing in that study found that metagenomes with higher polymorphic rates are more likely to contain multiple strains.

We performed a test to determine the expected dN/dS of mapping reads against an incorrect reference genome by using three species of *Fusoboacterium*, which were formerly considered subspecies of *F. nucleatum*: *F. nucleatum*, *F. polymorphum*, and *F. vincentii*. These genomes were run through prokka to generate gff files for polymut. The ANI of each genome compared to the other two was

determined with MASH[99] as part of the programme dRep[100] and found to be below the species-cutoff of 95% identity (Supplementary Fig. 16A). We used polymut.py to calculate the dN/dS for short read datasets of each genome mapped to the three reference genomes (Supplementary Fig. 16B–G), and took the average across all genomes, which was 1.94. We took this to be the expected dN/dS value when mapping a species against a closely-related but incorrect reference genome.

Nine simulated short-read datasets were generated from each of the three genomes with three read lengths and three different levels of deamination: long read length (100 bp), medium read length (75 bp), and short read length (50 bp), and no deamination, low deamination, and high deamination. These 9 short-read datasets were each mapped against all three reference genomes using bwa aln and the flags -n 0.01 -l 1024, and the script polymut.py was used to determine the number of synonymous SNPs, the number of non-synonymous SNPs, and the total number of sites compared, for each mapped bam file (Supplementary Fig. 16B–G). We found that the deamination level did not affect the dN/dS values for any of the species mapped to any of the others. Because the Pacific dataset reads have a short read length, we focused on the results of the short read length dataset, and calculated the average dN/dS for short reads across all deamination levels, which is 1.94. We took this to be the expected dN/dS value when mapping a species against a closely-related but incorrect reference genome.

For our samples, collapsed reads were mapped against the *Tannerella forsythia* genome (NC_016610.1) and the *Anaerolineaceae* bacterium oral taxon 439 genome (CP017039.1) using bwa aln and the flags -n 0.01 -l 1024 to account for ancient DNA damage. Mapped reads were masked according to their UDG treatment as described above (i.e., UDG-half treated reads were masked 1 bp on both ends, and non-UDG treated reads were masked 11 bp on each end). Masked mapped bam files were run through polymut with a minimum coverage requirement of 5X, min quality 30, and dominant frequency threshold of 0.8, and the gff files from NCBI RefSeq for the genome accessions above. We then calculated the ratio of non-synonymous SNPs to synonymous SNPs (dN/dS) for all samples mapped against the *Anaerolineaceae* bacterium oral taxon 439 or *Tannerella forsythia* genome (Supplementary Fig. 17).

## Eukaryotic DNA

In addition to analysing microbial DNA, we also investigated putative eukaryotic DNA within the samples. Identifying eukaryotic taxa within metagenomic datasets is challenging and requires multiple validation steps[23]. Additionally, it should be noted that some of the potential dietary items, such as kava (*Piper methysticum*) and the giant swamp taro (*Cyrtosperma merkusii*), from the Pacific may not be present in reference databases as they do not have sequenced nuclear or organelle genomes. We examined the MALT species table from profiling samples with the NCBI nt database to identify potential dietary hits. Within the well-preserved dental calculus samples, we identified DNA from five non-human eukaryotic species of interest: cattle (*Bos taurus*), dog (*Canis lupus familiaris*), broad fish tapeworm (*Dibothriocephalus latus*), bamboo (*Fargesia denudata*), and wheat (*Triticum aestivum*) (Supplementary Data 8). Other eukaryotic DNA present in the dataset belonged to commonly recognized contaminants, were assigned to genomes with known contamination[23], or represented highly unlikely taxa; these assignments were excluded from subsequent analysis. Data were next mapped (mapping quality set to 37) to the following genomes using nf-core/eager to produce damage patterns for authentication analysis: cattle (*Bos taurus*), dog (*Canis lupus familiaris*), broad fish tapeworm (*Dibothriocephalus latus*), bamboo (*Fargesia denudata*), and wheat (*Triticum aestivum*). For cattle, mapping was also performed to the zebu genome (*Bos indicus*), as this species may be a closer match for this region[101]. For bamboo, only the chloroplast genome was available. For wheat, the mapping was restricted to the mitochondrial

genome in this initial step, as the full wheat genome is very large (15.4 Gb). Damage patterns were investigated for individuals with at least 200 reads mapping to the specific taxa[23]. For taxa where 200 reads were not reached for any individuals, the damage patterns of the 10 individuals with the highest number of reads were investigated.

## Microparticle analysis

Fifty-four samples of decalcified dental calculus were analysed in this study. Phytoliths and starch granules were identified based on M.T.'s reference collection from several botanic gardens and herbariums in Vanuatu, New Zealand, Hawaii, and Rapa Nui, as well as other published material (for example[102]). Phytoliths were described based on the International Code for Phytolith Nomenclature (ICPN) 2.0[103]. Methods used for Rapa Nui[24,25] and Teouma[26] were previously published. All other samples had been decalcified in 0.5 M EDTA for aDNA extraction. After aDNA extraction, the remaining EDTA aliquot was rinsed in DDI water and a 40 μl drop containing most of the microparticle sample was placed on a glass slide and covered with a glass coverslip. Optical light microscopy was done with a Zeiss Axioscope (located in the Anthropology and Archaeology Department at the University of Otago). Each slide was examined in its entirety in vertical transects using cross-polarized and transmitted light to identify and photograph all microparticles. All microparticles were counted and separated into plant or animal types and then by morphotype. Identification of microparticles was done using published material as well as the Pacific-focused reference collection developed by M.T. The International Code for Phytolith Nomenclature 2.0 was used to name and describe all phytoliths[103].

**Efate**. The samples from Efate were not very microparticle rich; the most common microparticles were fungal spores and hyphae (commonly found in sediment samples and not possible to refine the ID) (Supplementary Fig. 14A). Sample EFE002.B did contain diatoms, but since it is only one sample and two diatoms, not much can be inferred from this. One starch granule was found in sample EFE003.B; however, it is round and less than 10 μm, which means it could come from just about any starch-containing plant (Supplementary Fig. 14B). A starch granule was also found in sample EFE006.B (Supplementary Fig. 14C). This granule is approximately 10 μm and angular; it is most similar to *Colocasia esculenta* or taro; however, it could also overlap with several other root crops and finding it in isolation makes it difficult to be certain.

**Futuna**. The Futuna samples had the highest number of starch granules of any location examined, and almost all were in one sample, FUT018.B. Of the 27 granules, 18 are ≤10 μm and so cannot be confidently identified to family/species. The remaining 9 granules correspond to Types 2 ($n=3$), 3 ($n=2$), 4 ($n=2$), 5 ($n=1$) and 8 ($n=1$) (Supplementary Fig. 15A, B, C, D, E)[45]. Type 2 granules could come from several root and tree crops and corn, which could be contamination, although the possibility that it is a genuine dietary component cannot be ruled out given that Europeans may have traded corn there as early as the 1600 s[104,105]. Type 3 granules could be from several root and tree crops. Type 4 granules are generally from *Tacca leontopetaloides* or Polynesian arrowroot–unfortunately, they also overlap with corn, so contamination cannot be ruled out. The type 5 granule could be from several root and tree crops. The type 8 granule is similar to one found in Lapita-aged samples from Vanuatu but could not be linked to any known reference species. A few different fungal spores and hyphae were also found in this sample. Sample FUT021.B contained one starch granule consistent with *Zea mays* (corn), which is likely contamination. There was also a piece of microcharcoal (Supplementary Fig. 15F) and three probable grass or Cyperaceae phytoliths (Supplementary Fig. 15G, H, I) in this sample.

**Taumako**. Almost all samples from Taumako contained fungal spores and/or hyphae, but not much else. Five starch granules were found in

three samples –all are either too small or damaged to be securely identified except for one, which is consistent with wheat starch and likely contamination (Supplementary Fig. 16A, B, C). Sample NMU116.A contained unusual microparticles that remain unidentified (Supplementary Fig. 16D); these may be associated with the high marine diet found in stable isotope results from the same population[106]. Finally, a few phytoliths were recovered; most were dentate elongate and likely from grass leaves; some were damaged and could not be adequately described (Supplementary Fig. 16E, F).

**Fiji**. There were quite a few phytoliths found in the Fiji samples, most of which are probably from grass leaves (blocky, elongate, bilobate and rondel types), along with a couple of samples that contained palm phytoliths (Supplementary Fig. 17A–J). These are all quite commonly found within the Pacific. There was one phytolith that could be species-specific that did not match anything in our reference database (Supplementary Fig. 17B); further reference samples would be needed to positively identify it, but it resembles phytoliths from the bark of species with medicinal properties in West Africa[107]. There are also two phytoliths that look like double-peaked glume rice phytoliths (Supplementary Figs. 17A, C), but they are very degraded so the resemblance may be an artefact. Several of the phytoliths appeared black or burnt (Supplementary Fig. 17G, H, I), which may be an indication of fire use. Overall, the Fijian samples were very mineral rich, likely due to insufficient removal of residual sediment for these samples (this issue is also noted in the aDNA methods above, where two samples had to be re-extracted and sequenced due to sediment contamination). Several of the mineral particles are large (probably too large to be inclusions in the dental calculus) and olive green (however, they are unlikely to be obsidian due to the lack of conchoidal fractures) (Supplementary Fig. 17K). Sample SIG044.A contained an unknown fibre probably of plant origin as there are no visible scales or a medulla; it may also be part of an insect (Supplementary Fig. 17L).

**Tongatapu**. Most samples from Tongatapu contained at least one phytolith. There were mostly palm phytoliths (spheroid echinate) (Supplementary Fig. 18B), followed by probable grass phytoliths (Supplementary Fig. 18A) and one instance of a Cyperaceae phytolith (Supplementary Fig. 18C). Two samples also contained sponge spicules (Supplementary Fig. 18D). One sample, TON001.C contained a unique looking fibre that may be a taphonomically damaged feather barbule (Supplementary Fig. 18E).

## Reporting summary

Further information on research design is available in the Nature Portfolio Reporting Summary linked to this article.

## Data availability

For detailed information about the archaeological sites and samples in this study, as well as research consultations and permissions, see Methods. The sequencing data for this study have been deposited in the European Nucleotide Archive under accession PRJEB61887 at https://www.ebi.ac.uk/ena/browser/view/PRJEB61887; individual accession numbers are provided in Supplementary Data 1. Published datasets analysed in this study are available in the European Nucleotide Archive; individual accession numbers are provided in Supplementary Data 2. Information about how this data was analysed is provided in Methods and Supplementary Methods. Source data are provided in the "06-publication" folder of https://github.com/ivelsko/pacific_calculus, also available through the https://doi.org/10.5281/zenodo.13903784.

## Code availability

The scripts used for analysis and figure generation are available at https://github.com/ivelsko/pacific_calculus, and through https://doi.org/10.5281/zenodo.13903784.

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

## Acknowledgements

We thank M. Abong of the Vanuatu Cultural Centre (VCC) for assistance during calculus sampling. Sampling of the Teouma dental calculus was done through a research agreement between M.T. and the Vanuatu National Cultural Council. The Teouma Archaeological Project is a joint initiative of the Vanuatu National Museum and the Australian National University (ANU), directed by M.S. and S.B. and at different times R. Regenvanu and M. Abong, both former Directors of the VCC. Funding for the project was provided by the Australian Research Council (DP 0556874 to S.B. and M.S.), the National Geographic Society (SRC 8038–06 to M.S.), the Pacific Biological Foundation (to M.S.), the Department of Archaeology and Natural History and School of Archaeology and Anthropology at the ANU (S.B. and M.S.), the Snowy Mountains Engineering Corporation Foundation (S.B. and M.S.) and Brian Powell (to S.B. and M.S.). The laboratory research and travel for excavation of the skeletal remains were funded by The Royal Society of New Zealand Marsden Fund (UOO0407 and 09-UOO-106 to H.R.B.) and a University of Otago Research Grant (to H.R.B). The support of the leaseholder M. R. Monvoisin and family is acknowledged, as is the support and assistance of the traditional landowners and population of Eratap Village. Detailed acknowledgement by the authors of the Vanuatu studies (S.B., H.R.B., J.F., R.S., M.S., E.W., and F.V.) are given in the published site reports for each location, and were supported by the Australian Research Council DP160103578 (to J.F., S.B., and F.V.). The excavation of the Pain Haka site in 2012 was funded by a grant from the Research Institute for Development (UMR Paloc to G.C.) and by additional funding from the French Embassy in Indonesia (to G.C. and T.S.), as well as a University of Otago Research Grant awarded to H.R.B. for the excavation and analysis of the human skeletal remains. Additional funding was provided by an Australian Research Council Discovery Grant (DP170100732 and DP200102872 to G.C.). Genetic data generation and analysis was supported by the Werner Siemens Stiftung ("Paleobiotechnology" to C.W.) and the Deutsche Forschungsgemeinschaft (DFG, German Research Foundation) under Germany's Excellence Strategy (EXC 2051, Project-ID 390713860 to C.W.). The Otago research presented here was funded by a University of Otago Doctoral Scholarship (to M.T.), a Royal Society of New Zealand Skinner Fund grant (to M.T.) and an Otago Centre for Electron Microscopy Student Research Award (to M.T.). We thank Alexander Hübner for discussions on strain analyses.

## Author contributions

Z.F., I.M.V., M.T., and C.W. designed and conceived the study. C.W. and I.M.V. supervised the project. A.T., S.B., H.R.B., G.C., J.D., J.F., J-C.G., R.K., C.M.L., E.M-S, K.N., A.T.O., C.P., A.B.R., R.S., T.S., M.S., A.T., F.V., and E.W. provided archaeological materials and resources. Z.F., M.T., and A.O. generated the genetic data. Z.F. performed initial data analysis. M.T. performed paleoethnobotanical data generation and data analysis. I.M.V. performed the final genetic data analysis. I.M.V. and C.W. wrote the manuscript; all authors edited the manuscript.

## Funding

## Competing interests

The authors declare no competing interests.

## Additional information

[1]Department of Archaeogenetics, Max Planck Institute for Evolutionary Anthropology, Leipzig, Germany. [2]Department of Archaeology, Max Planck Institute for Geoanthropology, Jena, Germany. [3]Southern Pacific Archaeological Research, Archaeology Programme, University of Otago, Dunedin, New Zealand. [4]Department of Anatomy, School of Biomedical Sciences, University of Otago, Dunedin, New Zealand. [5]Department of Archaeology and Natural History, College of Asia and the Pacific, The Australian National University, Canberra, Australia. [6]Department of Linguistic and Cultural Evolution, Max Planck Institute for Evolutionary Anthropology, Leipzig, Germany. [7]Department of Anthropology, Idaho State University, Pocatello, ID, USA. [8]Archaeology, School of Humanities, University of Sydney, Sydney, Australia. [9]Research Institute for Development and National Museum of Natural History, Paris, France. [10]BioArch South, Waitati, New Zealand. [11]Department of Anthropology, University of Oklahoma, Norman, OK, USA. [12]Department of Biological Sciences, Halmos College of Arts and Sciences, Nova Southeastern University, Fort Lauderdale, FL, USA. [13]Archaeo- and Palaeogenetics, Institute for Archaeological Sciences, Department of Geosciences, University of Tübingen, Tübingen, Germany. [14]Senckenberg Centre for Human Evolution and Palaeoenvironment at the University of Tübingen, Tübingen, Germany. [15]School of Biological Sciences, The University of Adelaide, Adelaide, Australia. [16]Vanuatu Cultural Centre, Port-Vila, Vanuatu. [17]National Research and Development Centre for Archaeology, Jakarta, Indonesia. [18]School of Archaeology and Anthropology, College of Arts & Social Sciences, The Australian National University, Canberra, Australia. [19]Direction de la Culture et du Patrimoine, Puna'auia, French Polynesia. [20]CNRS UMR 8068, MSH Mondes, Nanterre, France. [21]Faculty of Biological Sciences, Friedrich Schiller University, Jena, Germany. [22]Archaeogenetics Unit, Leibniz Institute for Infection Biology and Natural Products Research Hans Knoll Institute, Jena, Germany. [23]Department of Anthropology, Harvard University, Cambridge, MA, USA. [24]Present address: University of Copenhagen, Globe Institute, Copenhagen, Denmark. ✉e-mail: warinner@fas.harvard.edu

