## [Peer Review file · Nature Communications]

Exploring the potential of dental calculus to shed light on past human migrations in Oceania

Corresponding Author: Professor Christina Warinner

Version 1:

Reviewer comments:

Reviewer #1

(Remarks to the Author)

The manuscript from Velsko et al. presents the characterization of ancient oral microbiomes from populations of the Pacific Islands and Southeast Asia islands spanning over 3,000 years using ancient DNA high-throughput sequencing data obtained from archaeological dental calculi. The work is overall well-presented, and results and methods used are generally sounding. The authors propose in the title and introduction that the analysis of ancient dental calculus could shed light on past human migrations, and although I agree that this is an interesting hypothesis (and probably what inspired the study in first place), the results presented do not really shed light on this question. In that sense I propose changing the title (and intro), as it is misleading regarding what the manuscript actually contributes.

On the contrary, I rather found value in the manuscript at the following aspects: 1) It shows that there is a much better aDNA conservation in dental calculus than human DNA in tropical areas, thus reinforcing the value of this material for archaeogenetic studies, 2) It shows, using a much more extended aDNA dataset than in previous works, that the oral microbiome (at least the fraction that is recorded in calculus) is quite stable and homogenous across populations and time periods, 3) It also illustrates the complexity of dental calculus and oral microbiome data, by exemplifying how difficult it is to find drivers of microbial diversity and clear variation patterns between samples in relation to several variables, even those that a priori are expected to have an effect. Indeed, regarding the last point, several aDNA studies in dental calculus so far have tried (almost pushed) to find changes in the oral microbiome in response to a pre-existing working hypothesis (lifestyle, geographic location, etc.), while it is becoming more and more evident that the factors driving the ecology of the human oral microbiome are complex, and of different nature than those shaping the microbiome of other body sites, such as the gut microbiome.

In that regard, my humble suggestion—although this decision ultimately rests with the authors—is to redirect the focus of the work toward these aspects.

I have first some more general questions regarding the methods and analyses that are presented:

1) I think that some more precisions should be added in materials and methods about data processing and microbial diversity analyses. For example, how many sequences (or proportion of sequences) were taxonomically classified in the 73 well-preserved samples? Was the data from the “OTU tables” rarefied, normalized, or transformed for diversity analyses? Which distance metric was used in beta-diversity analyses? Also, to perform the Sourcetracker analyses, datasets were rarefied to 10,000 sequences. Isn't that too low to provide a reliable taxonomic profiling of the different samples? Which kind of bone was used for the “archaeological bones” category?

2) I always struggle to find a meaning in functional analyses of metagenomic data using KEGG annotations, so perhaps the authors could better explain or argue the relevance of the results obtained with this type of analysis, that a) could not be deduced from taxonomic data and b) has a biological meaning. Indeed, KEGG categories are often very broad and hard to interpret from a functional point of view. For example, what does it mean that cluster B (for which the only seemingly characteristic is that it does not contain Pacific samples) is enriched in KO “Protein families: genetic information processing”? The KEGG pathways within this category are also very broad, and supposedly present in every living cell, and some, like “Mitochondrial biogenesis” do not even make sense in the context of prokaryotic organisms. My point is that I feel that these results are often more conditioned by annotation and database biases (some taxa get better annotated than

others) than by truly metabolic or functional differences.

3) I think that some additional work is needed for the phylogenetic analyses. I would suggest following more closely some practices that are often used for the analysis of ancient pathogen genomes. For instance, NJ is, in my opinion, a less accurate phylogenetic method than ML or Bayesian, and in any case, tree topologies are often validated by more than one method (mostly when they are a substantial result of a manuscript) to confirm tree robustness. It is possible, though, that with the high level of heterozygosity in *Anaerolineaceae* bacterium oral taxon 439, ML or Bayesian methods will not be able to resolve the tree well, as it mixes in each branch or leaf the evolutionary histories of different strains present in each sample. In the context of phylogenetic analyses, and considering that filters were applied to eliminate heterozygous sites, low-covered SNVs, etc., I think it would be important to mention some stats from the strains included in the tree, such as, breadth of coverage, number/proportion of mapped and variant positions before and after applying filters, the size of the alignment used to build the tree, etc.

4) Regarding the inStrain analyses, I think they sort of show what is already observed in the phylogenetic tree, isn't it? For instance, AMH001 and AMH004 and all pairs sharing a popANI > 99.999 in Supp. Fig. 10 are evidently clustering together in the tree with very short branch lengths. Therefore, the additional information gained from the inStrain analyses should be more explicitly mentioned in the text, I believe.

5) I did not understand the rationale behind the use of dN/dS analyses (typically used to find genes or genomic regions under selection) as a strategy to infer if there are multiple closely-related species in the sample. I think a clear explanation about this is missing in both the main text and supplementary materials.

I have also some minor comments that I hereby list, more or less in order of appearance:

6) Line 174: I am not questioning the result, but I am surprised that only 19 out of 464 of ancient northern European samples yielded more than 5% of endogenous human DNA. Which kind of bones they were?

7) Lines 183-184: Changes in the oral microbiome between samples from different islands or time periods could be explained by many factors totally unrelated of migration, isn't it?

8) Line 185: Which metric was used for beta-diversity?

9) Lines 186-187: The sentence is not very clear. I only understood what was being referred by "number of clusters" after reading supplementary material.

10) Lines 187-189: what is the result or figure associated to this sentence?

11) Lines 198-199: The sentence is a bit unclear

12) Line 200-203: I don't see the point of this sentence. I find it anecdotal and uninformative.

13) Line 208: The tools and DBs used could be mentioned here.

14) Line 280: How were the three KO clusters and clusters A and B defined? Using also the clusGap method or some other? I could not find this defined in the text and I think it is necessary.

15) Line 285: Figure S7C shouldn't be replaced by Figure 4C?

16) Lines 286 – 289: Maybe if the analysis is repeated depleting first the *Ottowia* reads, it would become evident how much it was contributing to the observed clustering.

17) Figure 4D: The bars in the bar plot from figure 4D could be colored differently for cluster A and B, for clarity. Also, in the legend, it should be mentioned that the plot is aligned with Fig 4A (it could even be part of Figure 4A with the legend referred as bottom bar-plot panel).

18) Lines 307 and 309: "indicating that the reference strain used for mapping is closely related to the strains present in the samples". I don't see the link between the number of heterozygous sites and how close the strain/s is/are to the reference. Heterozygous sites can inform if there are more than one strain in the sample, but it does not about the genetic distance from the reference. Mapping parameters can be also tuned to favor only close related strains mapping to the reference (although I think the mapping parameters used are stringent enough, and if all strains are far from the reference genome, the problem is not solved either since no read would map).

19) Lines 313-316: Did you check if there is sign of recombination in *Anaerolineaceae* bacterium oral taxon 439 strains? Maybe masking them with Gubbins, plus eliminating heterozygous sites (as done already), can improve further the tree reconstruction.

20) Lines 352-353: The writing is strange here and could be corrected.

21) Lines 365-366: Isn't this phrase too speculative?

22) Lines 426 – 427: I did not follow the reasoning here. Why would a single phylogenetic cluster reflect that this was the last populated site? Also, the tree topologies of RapaNui samples are split in two clusters in both trees, and not necessarily grouping same samples, for those included in both trees, two results that I don't know how to interpret in the context of the phrase nor of migration histories mentioned in manuscript's title.

23) Lines 443: "affect the branching pattern within clades ". How is this manifested in the tree or in the results presented?

24) Line 621: The DB used for kraken2 could be mentioned here.

25) Line 660: Wouldn't this filter of 5X for SNVs be too high if strains (e.g., of *T. forsythia*) are kept when coverage is >2X ?

26) Lines 678: Which were the stats, filters or reasons, explaining the exclusion of the other 4 species?

27) Lines 705: It is not clear here that the eukaryotic sequences were analyzed with MALT on the Full NCBI NT database, and that these were the 5 relevant results obtained from it. In the way it is written, it seems that reads were only aligned against these 5 taxa. Also, out of curiosity, what is the advantage of annotating microbial taxa using a custom microbial RefSeq DB instead of the Full NCBI NT, given that the analysis with the latter was done anyway... (not every computer can run MALT on Full NCBI NT database, and I thought this was the main reason why people often chose microbial RefSeq DB)?

Reviewer #2

(Remarks to the Author)

I commend you for embarking on an ambitious attempt to do what no one has tried before, namely using archaeological dental calculus to study human migration. While your research is ground-breaking, it highlights that much more needs to be done before you can fully achieve your goal, including refining the sequencing methods to expand the reference genomic databases used for taxonomic profiling, increasing the ancient dental calculus dataset and including more geographic locations, and obtaining calculus samples from individuals with paired human genetic data. While discussing the potential of your work, you do a good job of conveying the limiting factors that affect the outcomes your study.

I only have a couple of comments:

The title of your manuscript leads the reader to expect a bit more light to be shed on migration by your study. Perhaps insert the word "potential." Exploring the potential of archaeogenetic studies of dental calculus to shed light on past human migrations in Oceania.

Page 15, Lines 425-427 – You state that the distinct single cluster potentially reflects that Rapa Nui was the last place to be colonized among the islands represented in this study. Perhaps you should mention that it is consistent with Rapa Nui being the last place to be colonized among the islands represented in this study, although the distinct cluster could also be interpreted as potentially due to a reduction in genetic diversity over time because of Rapa Nui's remoteness.

It would greatly enhance the article if a couple of the figures were clearer or larger.

Fig 1B - the modern and ancient calculus symbols are too close in color

Fig 4A – too small to see/read.

I enjoyed reading your article.

Reviewer #3

(Remarks to the Author)

In this study, the authors performed metagenomics analyses on archaeological dental samples obtained from 102 individuals from the Pacific/Southeast Asia region to determine the suitability of such samples for archaeogenetic studies on past human migrations in Oceania. They found that oral microbiome DNA preservation in calculus was higher than that of human DNA in archaeological bone and comparable to that of calculus from temperate regions. Additionally, the oral microbiome composition was distinctive compared to calculus from other regions (Europe, Africa, and Asia).

These findings, along with the other analyses conducted on bacterial species phylogeography, dietary DNA, and microparticles, demonstrate the significant potential of dental calculus to provide a substantial amount of information for the study of human populations and migrations in regions with climatic conditions that are unfavorable to DNA preservation. Therefore, I think that this work is interesting and likely to be significant in the field of paleobiology and human evolution in general.

The authors used the appropriate laboratory and bioinformatics methodologies for the analyses.

Minor specific comments:

The data generated from this study will serve as a crucial reference and comparison for future research projects.

Unfortunately, I was not able to find the ENA project PRJEB61887, thus I was not able to access the data to check it. I'm unsure if it will be made available after publication, or if there was an error with the provided accession number.

Regarding dietary DNA, have you examined whether there are any differences in dietary DNA and/or microparticles present in dental calculus between your samples and those from other regions? Do samples from other regions contain a higher quantity/quality of dietary DNA?

On line 122 and 169 of the manuscript and line 79 of the supplementary material, the total number of archaeological dental calculus analyzed is 103 and not 102 as in the rest of the work.

In Figure 2A, the legend includes the island of Rurutu, which is not referenced elsewhere in the manuscript.

Supplementary table S7 on microfossil results appears to be empty.

Version 2:

Reviewer comments:

Reviewer #1

(Remarks to the Author)

I have reviewed the revised version of the manuscript, and I am overall satisfied with authors' responses to my questions and with the modifications introduced to the article.

For some of the points that I had raised in my previous revision, I have few disagreements with authors' points of view or justifications, but these are just opinions (both, mine and theirs) and are thus beyond the principles of peer-review.

I have just a remaining comment/concern based on the new data and results that were incorporated:

- Supplementary figure 9: Despite the tests of concordance, when I see the trees presented in this new figure, mainly between masked and not masked trees, I really don't see them too similar, e.g., in plot D, or in C vs. A. Just to give some examples: in the tree in A, the most ancient nodes (purple circles) are internal nodes, while in figure C, they are mostly clustering apart in an independent cluster. How would this be interpreted? This is a significant change in tree topology that would affect the interpretations in terms of human migrations (e.g., Rapa Nui being basal in A and internal in C). While certainly less evident, there are also some discrepancies between the unmasked ML and NJ trees that are to some extent relevant in regards of human migrations, such as the cluster of S5,S2,S15,S7,S12,PAH003 and PAH002 that cluster together with SIG samples in ML and TON samples in NJ. Both trees would reflect quite different migration histories, for example, in the case of Efate, Flores and Vao.

Maybe using a tanglegram plot could make these discrepancies more visually evident. Overall, my point here is that under the principle of the article's title of shedding light on past human migrations, these trees are hard for me to interpret.

I agree that trees are rather consistent for *T. forsythia* in Supp. Fig 10, but it includes fewer number of populations (and most of those that change the most in Supp. Fig. 9 are missing in Supp. Fig 10) .

- I think that regarding the article's title and scope, the discussion lacks an explicit paragraph summarizing the evidence and different results presented in the work that contribute information about past human migrations. There are some sparse sentences mentioning this point, but since it is presented by the authors as a central point of the article, it should be more specifically discussed in my opinion.

Reviewer #2

(Remarks to the Author)

I am satisfied that the authors have adequately addressed my comments and appreciate the title change. Fig 1B would be enhanced if either of the overlapping symbols (ancient or modern) is changed to a contrasting color such as orange; however, I leave this up to the authors.

Reviewer #3

(Remarks to the Author)

The authors have addressed my comments.

Point-by-point response to reviewers

We received 38 specific comments from three reviewers. We present our responses to the reviewer comments below.

Reviewer #1 (Remarks to the Author):

Comment #1: *The manuscript from Velsko et al. presents the characterization of ancient oral microbiomes from populations of the Pacific Islands and Southeast Asia islands spanning over 3,000 years using ancient DNA high-throughput sequencing data obtained from archaeological dental calculi. The work is overall well-presented, and results and methods used are generally sounding. The authors propose in the title and introduction that the analysis of ancient dental calculus could shed light on past human migrations, and although I agree that this is an interesting hypothesis (and probably what inspired the study in first place), the results presented do not really shed light on this question. In that sense I propose changing the title (and intro), as it is misleading regarding what the manuscript actually contributes.*

On the contrary, I rather found value in the manuscript at the following aspects: 1) It shows that there is a much better aDNA conservation in dental calculus than human DNA in tropical areas, thus reinforcing the value of this material for archaeogenetic studies, 2) It shows, using a much more extended aDNA dataset than in previous works, that the oral microbiome (at least the fraction that is recorded in calculus) is quite stable and homogenous across populations and time periods, 3) It also illustrates the complexity of dental calculus and oral microbiome data, by exemplifying how difficult it is to find drivers of microbial diversity and clear variation patterns between samples in relation to several variables, even those that a priori are expected to have an effect. Indeed, regarding the last point, several aDNA studies in dental calculus so far have tried (almost pushed) to find changes in the oral microbiome in response to a pre-existing working hypothesis (lifestyle, geographic location, etc.), while it is becoming the more are more evident that the factors driving the ecology of the human oral microbiome are complex, and of different nature than those shaping the microbiome of other body sites, such as the gut microbiome.

In that regard, my humble suggestion—although this decision ultimately rests with the authors—is to redirect the focus of the work toward these aspects.

Response: We thank the reviewer for their kind assessment.

I have first some more general questions regarding the methods and analyses that are presented:

Comment #2a-e: *1) I think that some more precisions should be added in materials and methods about data processing and microbial diversity analyses. For example,*

- a) how many sequences (or proportion of sequences) were taxonomically classified in the 73 well-preserved samples?
- b) was the data from the “OTU tables” rarefied, normalized, or transformed for diversity analyses?
- c) Which distance metric was used in beta-diversity analyses?
- d) Also, to perform the Sourcetracker analyses, datasets were rarefied to 10,000 sequences. Isn't that too low to provide a reliable taxonomic profiling of the different samples?
- e) Which kind of bone was used for the “archaeological bones” category?

Responses: We have added clarification on these points as follows:

- a) The percent of classified reads for all samples is now included in a new column “Percent_assigned” in Tables S1 and S2.
- b,c) The species tables were center-log ratio transformed for diversity analyses, and all beta-diversity analyses were PCAs, which by default use Euclidean distance; this information has been added to the methods section.
- d) The SourceTracker tool was originally developed and tested on datasets rarefied to 1,000 sequences (Knights et al. 2011, doi:10.1038/nmeth.1650). We have further tested the limits of SourceTracker in other projects and found that increasing the number of reads beyond 5,000 does not substantially affect the proportion assigned to each source. There are small fluctuations in the proportions assigned to each source, but the source with the highest contribution is the least affected. In our case the strong oral signal in well-preserved samples remains the strongest signal even when read counts are down-sampled below 10,000. For our study, we chose to be conservative and use 10,000 sequences.
- e) The archaeological bones are published human femur bones from Mongolia, as well as human petrous bones from Taumako and Viti Levu, and now indicated in the Methods section “Preprocessing”.

Comment #3: *2) I always struggle to find a meaning in functional analyses of metagenomic data using KEGG annotations, so perhaps the authors could better explain or argue the relevance of the results obtained with this type of analysis, that a) could not be deduced from taxonomic data and b) has a biological meaning. Indeed, KEGG categories are often very broad and hard to interpret from a functional point of view. For example, what does it mean that cluster B (for which the only seemingly characteristic is that it does not contain Pacific samples) is enriched in KO “Protein families: genetic information processing”? The KEGG pathways within this category are also very broad, and supposedly present in every living cell, and some, like “Mitochondrial biogenesis” do not even make sense in the context of prokaryotic organisms. My point is that I feel that these results are often more conditioned by annotation and database biases (some taxa get better annotated than others) than by truly metabolic or functional differences.*

Response: We agree that it is difficult to interpret the results of microbial functional analyses using systems developed for eukaryotic organisms. The SEED classification of microbial metabolic pathways is the one we have found most useful; however, it is not included in the HUMANN pipeline. Nevertheless, we expect that functional differences may account for

taxonomic differences observed between sample groups, as taxonomic groups reflect coherent suites of metabolic functions. If organisms with similar metabolic processes are enriched in some sample groups compared to others, we expect those functions to be enriched, and perhaps more clearly or abundantly than individual taxa, as similar or related functions can be found in many species. It is disappointing that the category that was significantly different between groups in our comparison is one for which the name has little information, which is why we looked at which species were contributing the genes that were enriched in this group of functions.

We have added additional text to clarify this point: “Given the broadly general cellular processing categories included in this pathway, we assessed the genera that were contributing the orthologs in these pathways to see if we could glean more microbially-relevant information.”

Comment #4: 3) *I think that some additional work is needed for the phylogenetic analyses. I would suggest following more closely some practices that are often used for the analysis of ancient pathogen genomes. For instance, NJ is, in my opinion, a less accurate phylogenetic method than ML or Bayesian, and in any case, tree topologies are often validated by more than one method (mostly when they are a substantial result of a manuscript) to confirm tree robustness. It is possible, though, that with the high level of heterozygosity in Anaerolineaceae bacterium oral taxon 439, ML or bayesian methods will not be able to resolve the tree well, as it mixes in each branch or leaf the evolutionary histories of different strains present in each sample. In the context of phylogenetic analyses, and considering that filters were applied to eliminate heterozygous sites, low-covered SNVs, etc., I think it would be important to mention some stats from the strains included in the tree, such as, breadth of coverage, number/proportion of mapped and variant positions before and after applying filters, the size of the alignment used to build the tree, etc.*

Response: We have added additional phylogenetic trees built using maximum likelihood methods. We present these trees for both Anaerolineaceae bacterium oral taxon 439 and *T. forsythia* in the supplemental figures S9 and S10, respectively, along with tests for correspondence between the NJ and ML trees. We found high concordance between the trees built with different methods, and we see that the branching patterns with samples having higher proportions of heterozygous SNPs falling more basal in their respective clusters is consistent between methods as well. Mapping stats for *Anaerolineaceae* bacterium oral taxon 439 and *Tannerella forsythia* are presented in Supplementary Tables S4 and S5, respectively.

Comment #5: 4) *Regarding the inStrain analyses, I think they sort of show what is already observed in the phylogenetic tree, isn't it? For instance, AMH001 and AMH004 and all pairs sharing a popANI > 99.999 in Supp. Fig. 10 are evidently clustering together in the tree with very short branch lengths. Therefore, the additional information gained from the inStrain analyses should be more explicitly mentioned in the text, I believe.*

Response: We have added the following additional text to include inStrain more explicitly in the results: “The similarity of *Anaerolineaceae* bacterium oral taxon 439 strains (popANI > 99.99%)

in samples from the same islands, including Viti Levu (SIG), Taumako (NMU), and Rapa Nui (A*), is reflected in the close phylogenetic clustering of these samples. No identical *T. forsythia* strains were found between any samples (Supplemental Figure S11), but closely related strains (popANI > 99.91%) were found in samples from Rapa Nui (A*)."

Comment #6: 5) *I did not understand the rationale behind the use of dN/dS analyses (typically used to find genes or genomic regions under selection) as a strategy to infer if there are multiple closely-related species in the sample. I think a clear explanation about this is missing in both the main text and supplementary materials.*

Response: The reviewer is correct, traditionally dN/dS is used to infer selection between two genomes, comparing SNPs to a reference. However, we have used it for the application developed and published by Pasolli, et al. (2019, <https://doi.org/10.1016/j.cell.2019.01.001>), which was designed to use dN/dS to infer whether a metagenomic sample likely has multiple strains of a species. We have added additional text to the Supplemental methods in the section "Strain sharing" to explain the use of dN/dS here.

I have also some minor comments that I hereby list, more or less in order of appearance:

Comment #7: 6) *Line 174: I am not questioning the result, but I am surprised that only 19 out of 464 of ancient northern European samples yielded more than 5% of endogenous human DNA. Which kind of bones they were?*

Response: The samples are predominantly petrous bones, but also some dentin samples. These are all published values taken from supplemental tables of their corresponding publication.

Comment #8: 7) *Lines 183-184: Changes in the oral microbiome between samples from different islands or time periods could be explained by many factors totally unrelated of migration, isn't it?*

Response: It is true that there are other possible explanations, and we have changed this statement to "If present, such patterns might suggest that the calculus microbiome changed with human migration through ISEA and the Pacific, although specific factors affecting such changes would need elucidation".

Comment #9: 8) *Line 185: Which metric was used for beta-diversity?*

Response: All beta-diversity assessments were done with PCA, which by definition uses Euclidean distance. We added this information to the methods section. In addition, we have added a new analysis that assesses the variation in community structure, called FST-based Assessment of Variability across vectors of relative Abundances (FAVA), in contrast to measuring the similarity/difference between communities that is done by PCA (Figures 2C and 3D). FAVA confirmed that there is limited variability in species composition between samples

from Oceania and across the globe. The low variance across samples suggests that there are no major community structural shifts captured in the calculus samples we studied, as major shifts are expected to change the variability of a community as it adjusts and rebalances.

Comment #10: 9) Lines 186-187: *The sentence is not very clear. I only understood what was being referred by “number of clusters” after reading supplementary material.*

Response: We have clarified the reference to clusters with additional text: “We performed a beta-diversity analysis and visualized the samples using PCA (Figure 2A), and tested whether multiple clusters were present in the data. Testing the goodness of fit of sample cluster numbers to the data indicated that a single cluster optimally described the data (Supplemental Figure S2). This corresponded with the visual lack of distinct sample clustering in the PCA plot, in which the samples did not tightly cluster based on island or time period, nor did they plot along a cline that might suggest temporal or geographic change.”.

Comment #11: 10) Lines 187-189: *what is the result or figure associated to this sentence?*

Response: We have clarified the figure for this statement as part of the clarification for the point above: “This corresponded with the visual lack of distinct sample clustering in the PCA plot, in which the samples did not tightly cluster based on island or time period, nor did they plot along a cline that might suggest temporal or geographic change. ”

Comment #12: 11) Lines 198-199: *The sentence is a bit unclear*

Response: We have rearranged this and the preceding paragraph to clarify this point. Loss of AT-rich fragments in ancient dental calculus is an observation described in at least two publications. However, these studies did not explore how the loss of these fragments affects the abundance profile of species in relation to their GC content. Given the wide range of GC content in bacterial species that live in the mouth, it is possible that loss of AT-rich fragments will artificially reduce the relative abundance of species with low GC content, while artificially elevating the relative abundance of species with high GC content. This is an aspect of DNA preservation in ancient dental calculus that needs further exploration.

Comment #13: 12) Line 200-203: *I don’t see the point of this sentence. I find it anecdotal and uninformative.*

Response: This sentence allows for interpretation of the species communities in the context of microbiology physiology. We find that this is more insightful than a list of species names, which does not inherently imply specific metabolic and physiologic activity unless one is already familiar with these characteristics in the species being discussed. Oxygen tolerance is a fundamental aspect of bacterial physiology and a determinant of the kind of environment a species can survive in. Finding that there is a difference in the oxygen tolerance of species that characterize samples across the PCA provides insight into the biofilm biochemical characteristics of those samples, and the functional metabolism those species may be

performing. Whether a species is oxygen-tolerant or oxygen-intolerant is one of the first characteristics about a species that is determined, as it is essential to culture it in an atmosphere that will enable it to grow. The oxygen tolerance of species is recorded in reference manuals such as Bergeys Manual of Bacteriology, as well as in culture collection documentation, and is easy to look up.

The species loadings in PCA, which are taken from the output of the PCA directly, tell us how important a species is to locating a sample in PC space. Species with higher loadings have a stronger influence on the location of a sample in PC space. Here, the species with the strongest negative loadings in PC1 and the species with the strongest positive loadings in PC1 (Supplemental Table S3) have differing oxygen tolerances. This tells us that the species community in samples that plot in strong positive PC1 values have an oxygen-tolerant biofilm community, with most species being oxygen tolerant or oxygen-reliant. This corresponds to a younger, less mature community in *in vitro*, *in vivo* and *in situ* biofilm development studies, in which carbohydrate processing is common. The samples that plot in more negative PC1 values have a less oxygen-tolerant biofilm, with most species being oxygen-intolerant. This corresponds to a more mature biofilm community, in which protein degradation, ammonia production, and methane and sulfur reduction are common.

That this trend has been observed in multiple datasets supports that it is a true distinction in the community profiles of ancient dental calculus. It also indicates that if this is the major factor distinguishing ancient dental calculus samples, it may be very difficult to identify more subtle trends and distinctions in samples. Further work is needed to determine whether other signals or trends, such as any related to dietary or other cultural changes may be identified in such datasets.

Comment #14: 13) Line 208: The tools and DBs used could be mentioned here.

Response: We added this information.

Comment #15: 14) Line 280: How were the three KO clusters and clusters A and B defined? Using also the *clusGap* method or some other? I could not find this defined in the text and I think it is necessary.

Response: We selected these clusters based on the *clusGap* method, and have added this information to the supplemental methods section under the header Functional analysis. We have also added a new supplemental figure with the visualization of this clustering, Supplemental figure S6.

Comment #16: 15) Line 285: Figure S7C shouldn't be replaced by Figure 4C?

Response: We corrected this mistake.

Comment #17: 16) Lines 286 – 289: *Maybe if the analysis is repeated depleting first the Ottowia reads, it would become evident how much it was contributing to the observed clustering.*

Response: We were unable to extract the individual reads identified as *Ottowia* from the rma6 file produced by MALT, and so could not perform the functional analysis on a file with those reads removed. However, we removed all *Ottowia* hits from our species table and re-ran the PCA with this filtered table, and found that there is almost no difference in the sample loadings compared to the complete species table. This plot is added to Supplemental Figure 7 as panel F. This indicates that this species alone does not strongly shape the taxonomic profile of this dataset.

Comment #18: 17) Figure 4D: *The bars in the bar plot from figure 4D could be colored differently for cluster A and B, for clarity. Also, in the legend, it should be mentioned that the plot is aligned with Fig 4A (it could even be part of Figure 4A with the legend referred as bottom bar-plot panel).*

Response: We added a line to delineate clusters A and B in panel C, and clarified the legend as suggested. We found that changing the color of the bars did not make an impact because the bars are so low in cluster A that the color mostly isn't apparent.

Comment #19: 18) Lines 307 and 309: *“indicating that the reference strain used for mapping is closely related to the strains present in the samples”. I don't see the link between the number of heterozygous sites and how close the strain/s is/are to the reference. Heterozygous sites can inform if there are more than one strain in the sample, but it does not about the genetic distance from the reference. Mapping parameters can be also tuned to favor only close related strains mapping to the reference (although I think the mapping parameters used are stringent enough, and if all strains are far from the reference genome, the problem is not solved either since no read would map).*

Response: Our use of heterozygous here is indeed inappropriate, and we have changed it to multiallelic.

Comment #20: 19) Lines 313-316: *Did you check if there is sign of recombination in Anaerolineaceae bacterium oral taxon 439 strains? Maybe masking them with Gubbins, plus eliminating heterozygous sites (as done already), can improve further the tree reconstruction.*

Response: We ran Gubbins on the full alignment of the *Anaerolineaceae* genomes as well as on the full alignment of the *T. forsythia* genomes. For both there was low evidence of recombination, with 87-99% of bases in the clonal frame in *Anaerolineaceae* across the samples, and 88-100% of bases in the clonal frame in *T. forsythia* across the samples. We added the statistics from Gubbins as supplemental tables S6 and S7. Trees built using the masked alignment produced by Gubbins were plotted and included in supplemental figures S9-

S11. These trees were highly concordant with both the NJ and ML trees produced on the filtered SNP alignment (Kendall's W and mantel test shown in supplemental figures S9 and S10).

Comment #21: 20) Lines 352-353: *The writing is strange here and could be corrected.*

Response: We split this sentence into two to clarify as follows: “We were unable to identify any unambiguously positive evidence for dietary DNA in our samples, which may be due to the low number of non-microbial DNA sequences that were recovered. Alternatively, the sequences may be modern contaminants, or they may be aligning to an inaccurate reference genome²¹ (Supplemental Table S6).”

Comment #22: 21) Lines 365-366: *Isn't this phrase too speculative?*

Response: We have updated the sentence to be more clear about our reasoning: “This may indicate a greater reliance on starchy root crops and/or processing of plant foods than at Teouma where there were abundant phytoliths recovered from dental calculus. Compared to the published data from Rapa Nui, where numerous diatoms were found, the data presented here suggests better freshwater access on these islands than on Rapa Nui.”

Comment #23: 22) Lines 426 – 427: *I did not follow the reasoning here. Why would a single phylogenetic cluster reflect that this was the last populated site? Also, the tree topologies of RapaNui samples are split in two clusters in both trees, and not necessarily grouping same samples, for those included in both trees, two results that I don't know how to interpret in the context of the phrase nor of migration histories mentioned in manuscript's title.*

Response: We removed this statement and instead comment on the similarity of this pattern to other studies, which suggest that there are multiple lineages of these organisms. We speculate that the clusters on either side of the midpoint root may indicate distinct lineages of these organisms: “In phylogenetic trees for both *T. forsythia* and *Anaerolineaceae* bacterium oral taxon 439, the samples from Rapa Nui, the most remote island in this study, are in two clusters that fall on either side of the midpoint root. This pattern is similar to that of *Anaerolineaceae* bacterium taxon 439 and *T. forsythia* presented by Honap, et al (2023), and suggests that there are 2 distinct lineages of each species present in these samples. Alternatively, these lineages may instead be different species from those of the genomes used as the reference for mapping. However, due to the lack of additional reference genomes for oral *Anaerolineaceae* and *Tannerella*, this possibility is currently difficult to explore.”

Comment #24: 23) Lines 443: *“affect the branching pattern within clades “. How is this manifested in the tree or in the results presented?*

Response: We see that “Within a cluster of samples, there is a tendency for samples with higher levels of heterozygosity to fall basal to other samples (e.g., the cluster of samples from Taumako)”, as stated in the results. We see this pattern in the NJ and ML trees, which suggests that it is not an artifact of a specific tree building method.

Comment #25: 24) Line 621: The DB used for kraken2 could be mentioned here.

Response: There were multiple databases used for Kraken2, both of which are customized, and which are described in the supplemental text due to word limitations in the main text. As the data from the Kraken2 profiling with both databases is not used in the main text, we prefer to leave the information in the supplemental text to avoid confusion.

Comment #26: 25) Line 660: Wouldn't this filter of 5X for SNVs be too high if strains (e.g., of *T. forsythia*) are kept when coverage is >2X ?

Response: We used a minimum coverage cut-off of 5X on individual bases to have a reliable SNP call. The minimum whole-genome coverage for inclusion was 2X for *T. forsythia* because the overall coverage of this species was lower than for *Anaerolineaceae* bacterium oral taxon 439.

Comment #27: 26) Lines 678: Which were the stats, filters or reasons, explaining the exclusion of the other 4 species?

Response: We moved references to the other 4 species to the Supplemental Methods to avoid confusion. We chose to generate trees from *Anaerolineaceae* bacterium oral taxon 439 and *Tannerella forsythia* because these have been used in other ancient dental calculus studies and therefore contribute to a growing body of knowledge around these taxa in historic and ancient oral samples. Only *Anaerolineaceae* bacterium oral taxon 439 and *Tannerella forsythia* had sufficient genome breadth and depth of coverage across multiple samples for reliable strain comparison with inStrain.

Comment #28: 27) Lines 705: It is not clear here that the eukaryotic sequences were analyzed with MALT on the Full NCBI NT database, and that these were the 5 relevant results obtained from it. In the way it is written, it seems that reads were only aligned against these 5 taxa. Also, out of curiosity, what is the advantage of annotating microbial taxa using a custom microbial RefSeq DB instead of the Full NCBI NT, given that the analysis with the latter was done anyway... (not every computer can run MALT on Full NCBI NT database, and I thought this was the main reason why people often chose microbial RefSeq DB)?

Response: We now provide a more detailed explanation in the supplemental material for describing how we identified and analyzed potential dietary taxa: "nf-core/eager was used to align the DNA sequences to reference genomes (Supplemental Table S6) for the five species of interest (with mapping quality set to 37), which were selected from the output of dataset mapping with MALT against the NCBI nt database (See Supplemental Methods for details):...".

We chose to use a custom RefSeq database for analysis of the oral microbial content of the dental calculus rather than the NCBI nt database because we wanted to compare our data to

published datasets and to process these datasets in the same way. As MALT is a memory-intensive and time-intensive program to run, it was more economical for us to profile these datasets with the custom RefSeq database rather than profile all comparative datasets with the NCBI nt database. However, for the dietary analysis of our samples we used the NCBI nt database to allow detection of plant and animal taxa without fully sequenced genomes.

Reviewer #2 (Remarks to the Author):

Comment #29: *I commend you for embarking on an ambitious attempt to do what no one has tried before, namely using archaeological dental calculus to study human migration. While your research is ground-breaking, it highlights that much more needs to be done before you can fully achieve your goal, including refining the sequencing methods to expand the reference genomic databases used for taxonomic profiling, increasing the ancient dental calculus dataset and including more geographic locations, and obtaining calculus samples from individuals with paired human genetic data. While discussing the potential of your work, you do a good job of conveying the limiting factors that affect the outcomes your study.*

Response: We thank the reviewer for their kind assessment.

I only have a couple of comments:

Comment #30: *The title of your manuscript leads the reader to expect a bit more light to be shed on migration by your study. Perhaps insert the word “potential.” Exploring the potential of archaeogenetic studies of dental calculus to shed light on past human migrations in Oceania.*

Response: Thank you - that’s a fair point. We have updated the title to: Exploring the potential of archaeogenetic studies of dental calculus to shed light on past human migrations in Oceania

Comment #31: *Page 15, Lines 425-427 – You state that the distinct single cluster potentially reflects that Rapa Nui was the last place to be colonized among the islands represented in this study. Perhaps you should mention that it is consistent with Rapa Nui being the last place to be colonized among the islands represented in this study, although the distinct cluster could also be interpreted as potentially due to a reduction in genetic diversity over time because of Rapa Nui’s remoteness.*

Response: When averaging the number of SNPs for the Rapa Nui samples together and all other samples together, there are more SNPs in the Rapa Nui samples than in the others, for both species, so there doesn’t seem to be lower diversity in those samples. We recognize that our original sentence was unclear, so we removed this statement and instead focus on the similarity of this pattern to other studies, which suggests that there are multiple lineages of these organisms. We speculate that the clusters on either side of the midpoint root may indicate distinct lineages of these organisms: “In phylogenetic trees for both *T. forsythia* and *Anaerolineaceae* bacterium oral taxon 439, the samples from Rapa Nui, the most remote island in this study, are in two clusters that fall on either side of the midpoint root. This pattern is similar

to that of *Anaerolineaceae* bacterium taxon 439 and *T. forsythia* presented by Honap, et al, and suggests that there are 2 distinct lineages of each species present in these samples. Alternatively, these lineages may instead be different species from those of the genomes used as the reference for mapping. However, due to the lack of additional reference genomes for oral *Anaerolineaceae* and *Tannerella*, this possibility is currently difficult to explore.”

Comment #32: *It would greatly enhance the article if a couple of the figures were clearer or larger.*

Fig 1B - the modern and ancient calculus symbols are too close in color

Fig 4A – too small to see/read.

Response: We changed the shape of the other ancient calculus and the Pacific calculus to distinguish them more clearly from modern calculus, and increased the size of figure 4.

I enjoyed reading your article.

Reviewer #3 (Remarks to the Author):

Comment #33: *In this study, the authors performed metagenomics analyses on archaeological dental samples obtained from 102 individuals from the Pacific/Southeast Asia region to determine the suitability of such samples for archaeogenetic studies on past human migrations in Oceania. They found that oral microbiome DNA preservation in calculus was higher than that of human DNA in archaeological bone and comparable to that of calculus from temperate regions. Additionally, the oral microbiome composition was distinctive compared to calculus from other regions (Europe, Africa, and Asia).*

These findings, along with the other analyses conducted on bacterial species phylogeography, dietary DNA, and microparticles, demonstrate the significant potential of dental calculus to provide a substantial amount of information for the study of human populations and migrations in regions with climatic conditions that are unfavorable to DNA preservation. Therefore, I think that this work is interesting and likely to be significant in the field of paleobiology and human evolution in general.

The authors used the appropriate laboratory and bioinformatics methodologies for the analyses.

Response: We thank the reviewer for their kind assessment.

Minor specific comments:

Comment #34: *The data generated from this study will serve as a crucial reference and comparison for future research projects. Unfortunately, I was not able to find the ENA project PRJEB61887, thus I was not able to access the data to check it. I'm unsure if it will be made available after publication, or if there was an error with the provided accession number.*

Response: The data are uploaded to ENA, but are not yet publicly available. We keep the data private until acceptance of the manuscript, and then will release it. Since ENA does not offer a

reviewer link like the NCBI SRA, we have provided a copy of the upload confirmation. All data is paired-end (R1/R2, not merged), adapter-removed and quality-trimmed, with human reads removed.

Comment #35: *Regarding dietary DNA, have you examined whether there are any differences in dietary DNA and/or microparticles present in dental calculus between your samples and those from other regions? Do samples from other regions contain a higher quantity/quality of dietary DNA?*

Response: We have not done any systematic comparison of dietary DNA in calculus samples from other regions. In general, we have found that in ancient dental calculus there are very few reads that can be reliably attributed to a dietary source, and we have not focused on this aspect of calculus DNA composition in our previous studies. The presence of microparticles is also highly variable across calculus samples, and we are not aware of any studies that have compared the quantity/quality of microparticle detection across regions.

Comment #36: *On line 122 and 169 of the manuscript and line 79 of the supplementary material, the total number of archaeological dental calculus analyzed is 103 and not 102 as in the rest of the work.*

Response: We have corrected this mistake.

Comment #37: *In Figure 2A, the legend includes the island of Rurutu, which is not referenced elsewhere in the manuscript.*

Response: We have removed this from the legend.

Comment #38: *Supplementary table S7 on microfossil results appears to be empty.*

Response: We have corrected this oversight, and the microfossil data is now included in the supplemental tables, with the updated table number S9.

Point-by-point response to reviewers

We received 3 specific comments from two reviewers. We present our responses to the reviewer comments below.

Reviewer #1 (Remarks to the Author):

Comment #1: *I have reviewed the revised version of the manuscript, and I am overall satisfied with authors' responses to my questions and with the modifications introduced to the article.*

For some of the points that I had raised in my previous revision, I have few disagreements with authors' points of view or justifications, but these are just opinions (both, mines and theirs) and are thus beyond the principles of peer-review.

I have just a remaining comment/concern based on the new data and results that were incorporated:

- Supplementary figure 9: Despite the tests of concordance, when I see the trees presented in this new figure, mainly between masked and not masked trees, I really don't see them too similar, e.g., in plot D, or in C vs. A. Just to give some examples: in the tree in A, the most ancient nodes (purple circles) are internal nodes, while in figure C, they are mostly clustering apart in an independent cluster. How would this be interpreted? This is a significant change in tree topology that would affect the interpretations in terms of human migrations (e.g., Rapa Nui being basal in A and internal in C). While certainly less evident, there are also some discrepancies between the unmasked ML and NJ trees that are to some extent relevant in regards of human migrations, such as the cluster of S5,S2,S15,S7,S12,PAH003 and PAH002 that cluster together with SIG samples in ML and TON samples in NJ. Both trees would reflect quite different migration histories, for example, in the case of Efate, Flores and Vao. Maybe using a tanglegram plot could make these discrepancies more visually evident. Overall, my point here is that under the principle of the article's title of shedding light on past human migrations, these trees are hard for me to interpret.

*I agree that trees are rather consistent for *T. forsythia* in Supp. Fig 10, but it includes fewer number of populations (and most of those that change the most in Supp. Fig. 9 are missing in Supp. Fig 10) .*

Response: We agree that the trees do show differences, but this in part due to the fact that they are midpoint rooted with no explicit outgroup included in the phylogenetic analysis, and so some differences are expected. Unfortunately, we are unable to provide a suitable outgroup for this analysis due to a lack of other reference genomes within Anaerolinaceae. This further emphasizes the need for more concerted research on this important family of oral bacteria. We agree that some of the tree branching patterns are unstable and difficult to interpret, and we believe this is because the input data does not represent individual genomes, but rather

chimeric genomes produced by the mapping of multiple species to a single reference genome (because no other oral isolate reference genomes are available for this family). As such, we intentionally avoid interpreting the branching patterns in the context of human migration because we believe the tree topology may be affected by this chimerism artifact, which represents the limits of current approaches. We have added four sentences to the results section to make these issues clearer and more explicit.

We believe a major contribution of our study is to document in detail the strengths and shortcomings of a mapping based approach for migration reconstruction, especially as this is the only method used so far for these analyses. For problematic taxa, we believe this problem may be overcome by pursuing a radically different approach involving ultra-deep sequencing of these metagenomes and performing de novo metagenomic assembly to separate the multiple species/strains of Anaerolinaceae so that more robust trees can be created. We are actively developing the methods to do this in a follow-up study. We hope we have made it clearer why we do not focus more on interpreting the mapping-based trees. We do not feel that the current Anaerolinaceae tree topologies reliably reflect human migration histories, although they do likely reflect patterns of dominant strain sharing between islands.

Comment #2: *I think that regarding the article's title and scope, the discussion lacks an explicit paragraph summarizing the evidence and different results presented in the work that contribute information about past human migrations. There are some sparse sentences mentioning this point, but since it is presented by the authors as a central point of the article, it should be more specifically discussed in my opinion.*

Response: We see great promise in tracing human migration histories through oral microbiome genomes, but show that reference genome mapping approaches are insufficient (to differing degrees dependent on species) to achieve this aim. We have detailed the specific problems with this approach using specific case studies and make recommendations for how these challenges may be overcome using de novo metagenomic assembly of oral microbial genomes. We believe our final paragraph makes these points clear and we are reluctant to add additional interpretive text that may introduce more speculative statements. We appreciate the reviewer's comments and feedback but wish to keep our discussion in its current form.

Reviewer #2 (Remarks to the Author):

I am satisfied that the authors have adequately addressed my comments and appreciate the title change. Fig 1B would be enhanced if either of the overlapping symbols (ancient or modern) is changed to a contrasting color such as orange; however, I leave this up to the authors.

Response: We have updated the color of the symbol for modern dental calculus to show greater contrast to ancient dental calculus in Figure 1B.